# Biocatalysed synthesis planning using data-driven learning

Daniel Probst [1,2✉], Matteo Manica [1], Yves Gaetan Nana Teukam [1], Alessandro Castrogiovanni[1,2], Federico Paratore [1] & Teodoro Laino[1,2]

Enzyme catalysts are an integral part of green chemistry strategies towards a more sustainable and resource-efficient chemical synthesis. However, the use of biocatalysed reactions in retrosynthetic planning clashes with the difficulties in predicting the enzymatic activity on unreported substrates and enzyme-specific stereo- and regioselectivity. As of now, only rule-based systems support retrosynthetic planning using biocatalysis, while initial data-driven approaches are limited to forward predictions. Here, we extend the data-driven forward reaction as well as retrosynthetic pathway prediction models based on the Molecular Transformer architecture to biocatalysis. The enzymatic knowledge is learned from an extensive data set of publicly available biochemical reactions with the aid of a new class token scheme based on the enzyme commission classification number, which captures catalysis patterns among different enzymes belonging to the same hierarchy. The forward reaction prediction model (top-1 accuracy of 49.6%), the retrosynthetic pathway (top-1 single-step round-trip accuracy of 39.6%) and the curated data set are made publicly available to facilitate the adoption of enzymatic catalysis in the design of greener chemistry processes.

[1] IBM Research Europe, CH-8803 Rüschlikon, Switzerland. [2] National Center for Competence in Research-Catalysis (NCCR-Catalysis), Rüschlikon, Switzerland. ✉email: dpr@zurich.ibm.com

Chemistry fostered the unprecedented rise of overall human wealth and well-being during the past two centuries, and today it is our trump card for averting and mitigating global crisis while reshaping our lives towards a more responsible use of natural resources[1,2]. Innovation in synthetic chemistry will be critical in making chemical processes and products more sustainable, resource-efficient and CO2-neutral[3]. While the design and development of catalysts are at the heart of greening chemistry, biocatalysis, together with chemoinformatics and artificial intelligence, can already accelerate the adoption of existing sustainable catalytic processes[4].

At the core of biocatalysis are enzymes, an integral part of all living organisms used in important industrial processes thanks to the multiple key advantages over conventional chemical reagents. In addition to their extremely high catalytic activity, enzymes catalyse chemo-, regio- and stereo-selective reactions and are both reusable and allow for an easy recovery of products when immobilised[5]. A further advantage of enzyme-catalysed reactions is that they are usually performed in water under mild conditions and thus significantly reduce waste. Moreover, enzymes themselves are fully degradable in the environment, and as such, they represent an important strategy towards greener chemistry[6]. Therefore, it is not surprising that enzymes are one of the key enablers of sustainable chemical processes[7,8], with a growing interest in their use to convert waste into valuable raw materials at an industrial scale[9]. Although the ability to use enzymes as catalysts in the organic synthesis of chemical compounds gained widespread attention for large-scale production[10–12], enzymes are still far from being widely adopted in daily synthetic laboratory works. The narrow substrate scope available from enzymatic databases, the difficulties in identifying patterns within enzymes classes that would extend the range of their applicability to unreported substrates, and the distinct stereo- and/or regioselectivities are domain-specific knowledge factors that make the adoption of enzymatic processes a challenging problem for synthetic chemists[13].

The knowledge gap between large corpora of enzymatic chemical reactions data and the human understanding of the structure-activity relationship hinder the ability to predict successful routes[14–16] when the substrates of interests are not directly associated with an enzyme. Machine learning and data-driven approaches may be useful strategies to capture the hidden patterns in large enzymatic data sets, similar to the proven ability to learn chemical reaction patterns from complex chemistry knowledge collections[17]. The extraction of chemical reaction rules from large data sets of traditional organic chemistry reactions[18] is one of the most successful examples[19] of providing transparency and explainability through AI applications in chemistry to date.

Despite the impact on traditional synthetic organic chemistry, computer-aided synthesis planning tools using biocatalytic reactions are in the early days of their development. Currently, only rule-based methods for predicting biosynthesis pathways have been examined, such as the ATLAS of Biochemistry or RetroRules[20–22]. Lately, RetroBioCat[23] became the first chemoinformatic approach for easing the adoption of biocatalytic reactions specifically for chemical synthesis. However, the implementation relies on a set of expertly encoded reaction rules coupled with a system for retrieving database records to enable the use of biocatalysis in synthetic organic chemistry. This use of reaction templates slows down the curation of newly collected data, requiring the intervention of human experts, and suffers from the limitation in capturing the effects on the reaction centre of long-range substituents. Shortly after, Kreutter et al.[24] presented a forward reaction prediction model based on the Molecular Transformer[25]. This approach exploits a multitask transfer learning to train a Molecular Transformer architecture, originally trained with chemical reactions from the US Patent Office (USPTO) data set, with 32,000 enzymatic transformations, each one annotated with the corresponding enzyme name. The enzymatic transformer model predicts the products formed from a given substrate and enzyme in the forward prediction task, reaching an accuracy of 54% when using the enzyme name information only and 62% when using the complete enzyme information as a full sentence (often also including the organism name). The approach addresses some of the concerns around scalability and data curation of reaction templates. However, the use of enzyme names as reaction tokens adds an additional level of challenge when trying to learn chemical reactivity patterns among enzymes with different names but belonging to closely related families. In addition, the lack of a corresponding backward model in this work does not allow for retrosynthetic planning.

Here, we generalise the use of the Molecular Transformer by adopting a tokenisation system based on enzyme classes and introducing an extension of the retrosynthetic algorithm by Schwaller et al.[26] to biocatalysis. Compared to the previous work by Kreutter et al., a backward model allows to predict substrates and enzyme classes given a target product, enabling retrosynthetic pathway prediction. In addition, we incorporate the EC (enzyme commission) number into the reaction SMILES, rather than encoding enzymes with their natural language name. Enzymatic reactions and the accompanying EC numbers were extracted from four databases, namely Rhea, BRENDA, PathBank and MetaNetX and merged into a new data set, named ECREACT, containing enzyme-catalysed reactions with the respective EC number, as shown in Fig. 1.

The forward prediction model achieves an accuracy of 49.6%, 63.5% and 68.8% top-1 and top-5, and top-10 respectively, while the single-step retrosynthetic model shows a round-trip accuracy of 39.6%, 42.3% and 42.6%, top-1, top-5 and top-10, respectively.

## Results

**Data set**. The enzymatic reaction data set with related EC (enzyme commission) numbers was created by merging entries extracted from Rhea ($n = 8659$), BRENDA ($n = 11130$), PathBank ($n = 31047$) and MetaNetX ($n = 34485$)[27–30]. This data set was then further processed by (1) removing products that occur as reactants in the same reaction, (2) removing known co-enzymes

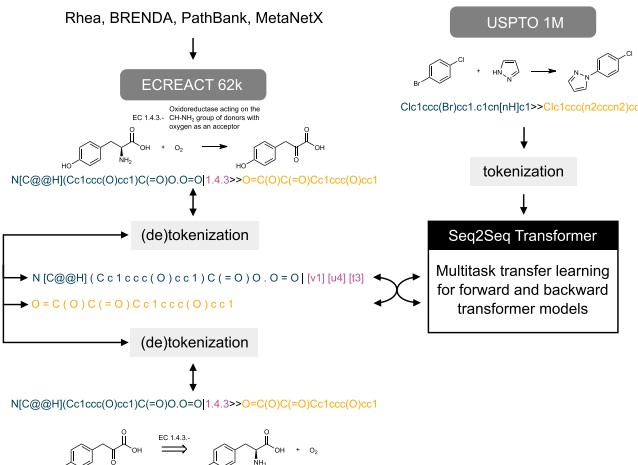

**Fig. 1 Introducing enzymes as green catalysts to data-driven template-free chemical synthesis.** The molecular transformer was trained on chemical reactions extracted from the USPTO data set and the new ECREACT data set using multitask transfer learning.

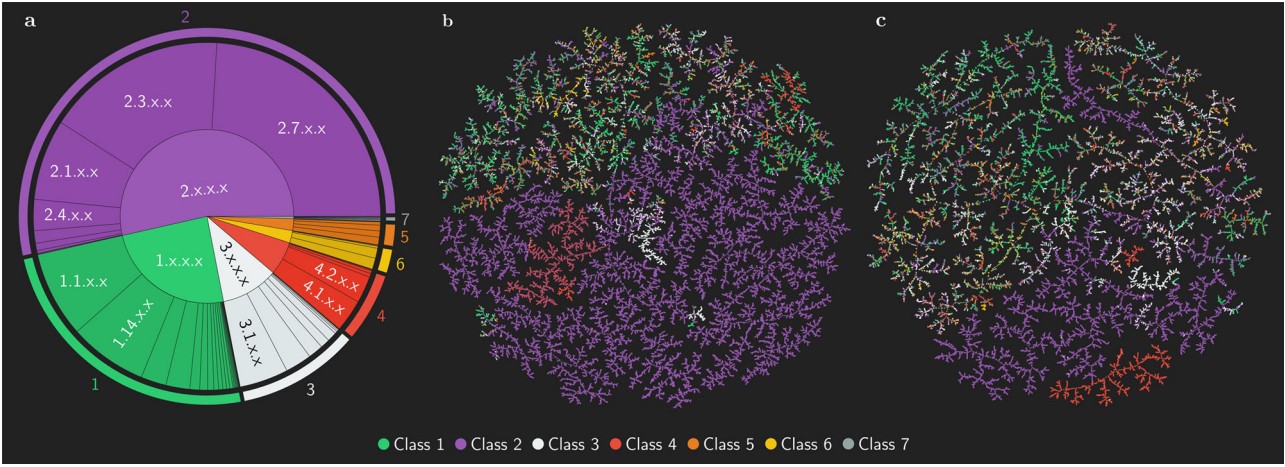

**Fig. 2 Enzyme class, substrate and product distributions of the data set ECREACT. a** The distribution of samples at EC-levels 1 (corresponding to enzyme classes) and 2 (corresponding to enzyme sub-classes) for oxidoreductases (class 1), transferases (class 2), hydrolases (class 3), lyases (class 4), isomerases (class 5), ligases (class 6) and translocases (class 7), in the ECREACT EC3 data set. A more extensive visualisation of the distribution of EC-levels 1, 2 and 3 can be found in Supplementary Fig. 3. TMAPs visualising the distribution of MAP4-encoded (**b**) reactants and (**c**) products in the ECREACT EC3 subset coloured by enzyme class corresponding to EC-level 1. Distributions of molecular distances (MAP4) per class are shown in Supplementary Fig. 1. While molecules of transferase- (class 2), lyase- (class 4), and, to a lesser extent, hydrolase-catalysed (class 3) reactions populate regions of the chemical space specific to each class (homogeneous), molecules from other classes are found in predominantly heterogeneous regions.

and common byproducts from the products in reactions that exceed one product (Supplementary Tables 1 and 2), (3) removing molecules with a heavy atom count < 4 from the products and (4) removing reactions with >1 or <1 products or no reactants. The resulting data set contains 62,222 unique reaction–EC number combinations. The data set is available in 5 different token schemes: With no EC number (EC0, $n = 55115$), only EC-level 1 (EC1, $n = 55707$), EC-levels 1-2 (EC2, $n = 56222$), EC-levels 1-3 (EC3, $n = 56579$) and EC-levels 1-4 (EC4, $n = 62222$). The different token schemes result in different set sizes as the removal of EC-levels leads to duplication and removal of extended reaction SMILES. Given the low specificity of enzyme information in EC1 and EC2 tokens, and the insufficient sampling for EC4, which is often confined to one enzyme-substrate example only, the EC3 data set remained the only one containing sufficient variability in terms of enzyme-substrate examples across individual tokens. Figure 2a shows the composition of the data set with token scheme EC3, containing 62,403 unique enzymatic reactions. At EC-level 1, which corresponds to enzyme classes, EC 2.x.x.x (transferases) account for 53.5% of total entries, EC 1.x.x.x (oxidoreductases) for 24.5%, EC 3.x.x.x (hydrolases) for 10.7%, EC 4.x.x.x (lyases) for 6.3%, EC 6.x.x.x (ligases) for 2.3%, EC 5.x.x.x (isomerases) for 2.2% and EC 7.x.x.x (translocases) for 0.4%. The high fraction of transferase-catalysed reactions is a consequence of the large number of non-primary lipid pathways stored in Path-Bank. Among transferases, the most common subclasses at EC-level 2 are EC 2.7.x.x (transferases transferring phosphorus-containing groups) at 24.5%, EC 2.3.x.x (acetyltransferases) at 16.8% and EC 2.1.x.x (transferases transferring one-carbon groups) 7.5%. The complete information on the distribution of samples across EC-levels 2 and 3 is provided in the supplementary information (Supplementary Tables 3 and 4, with a breakdown of the data set by data source shown in Supplementary Fig. 2).

The distribution of the available data reveals a heavy imbalance in the distribution of the enzyme-substrate examples. Whereas transferase-catalysed reactions encompass few subclasses at EC-level 3 with large sample size, the oxireductase- and hydrolase-catalysed reactions are divided into many subclasses with a small sample size at EC-level 3. Although lyases, isomerases, ligases and translocases are split into fewer subclasses at EC-level 3, most of

them contain very few samples. Therefore, the evaluation of the performance of the data-driven models will need to consider the different populations of each EC-level 3 subclass for a proper assessment.

A further property of interest regarding the reaction is the distribution of reactants and products within and across the enzyme classes at EC-level 1. The data set created with the EC3 token scheme contains 141,051 (56,017 unique) reactants and 62,403 (53,658 unique) products. Given the nature of the data sources, most reactants and products are metabolites. In Fig. 2b, c, we show the distribution of the compounds in the substrate and product chemical spaces using the 2048-dimensional binary MAP4 fingerprints and embedding them using TMAP[31,32]. The data points, coloured by enzyme class corresponding to EC-level 1, highlight the different distributions within the substrate set (containing cofactors) and the product set (where co-enzymes and common byproducts have been removed). Substrates and products of transferase- (class 2), lyase- (class 4) and, to a lesser extent, hydrolase-catalysed (class 3) reactions populate regions of the chemical space specific to each class (homogeneous), with little overlap with other classes. The chemical space covered by the molecules belonging to the remaining classes is non-specific (heterogeneous), with wide areas shared among different classes. The location of the substrates and products in homogeneous regions acts as an implicit feature, reducing the importance of the EC number token (explicit feature) during training. The lack of implicit features in substrates and products belonging to heterogeneous regions requires the use of explicit tokens (EC numbers) during training to learn the chemical transformation rules.

**Model selection.** A forward and backward model was trained for each of the ECREACT token schemes, with EC0 acting as a control on the influence of including enzymatic information in the reaction. The trained models were evaluated for forward, backward, round-trip and EC number prediction accuracy using 5% test splits with the condition that a product in the test split must not occur as a product in the training split (Fig. 3). The results show that the EC3 token scheme has a better forward performance than EC0 and EC2, yet performs slightly worse than EC1 and EC4 (Fig. 3a). In the backward prediction task, EC3 performs slightly worse than EC0, EC1

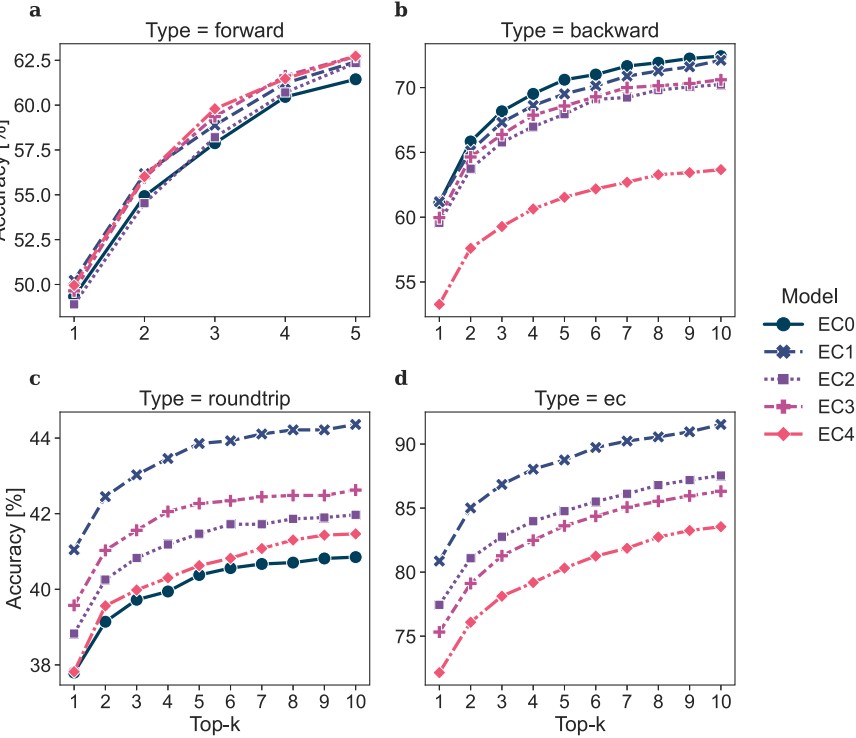

**Fig. 3 Overall accuracies of models based on different ECREACT token schemes EC0, EC1, EC2, EC3 and EC4.** Accuracies are reported for **a** forward prediction, **b** backward prediction, **c** round-trip prediction (a forward prediction followed by a backward prediction) and **d** backward EC number only prediction. Top-n indicates the accuracy when checking the top n predictions for the correct one.

and EC2, but significantly better than EC4; this is most likely due to the low number of samples in each EC4 category (Fig. 3b and Supplementary Table 5). Regarding the round-trip accuracy[26], EC3 performs better than both EC2 and EC4 (Fig. 3c). When solely focusing on the prediction of the correct EC number in the backward prediction, the models perform better the less detailed information they have to predict (Fig. 3d). These data show that the inclusion of enzymatic information in the form of the EC number does not affect the prediction performance negatively as long as each EC category has a sufficient number of training samples, which restricts the use of EC4 (Supplementary Fig. 16). The EC1 token, although performing well across different metrics, averages across reaction classes with different schemes and is for this reason of little interest for retrosynthetic purposes. The EC3 token scheme balances specificity of enzyme information with performance compared to the other ECREACT token schemes, resulting in a prediction performance similar to or better than EC1 and EC2, while retaining detailed information of the reaction-specific enzyme. Therefore, the relative performance among the five ECREACT token schemes EC0, EC1, EC2, EC3 and EC4, identifies EC3 as the one with the richest amount of statistically significant information.

**Forward prediction**. The Dataset was constructed following the details reported in the Methods, Data Sets and Model Training. We split the EC3 data set ($n = 56,579$) into a test and a training set, enforcing a zero overlap between the product distributions of the two ensembles, i.e. no product molecule present in the test set appears in the training set. Although this splitting penalises the measure of the performance of the forward model when compared to random splitting[25,26], it prevents the evaluation of the forward and backward predictions to be affected by memorisation of reaction records rather than by learning enzyme-substrate patterns. Despite the various similarities between the use of EC number and the use of catalyst tokens in chemical reactions, we assessed the learning of

biocatalytic signals by randomising the EC numbers in the test set within and across classes (corresponding to EC-level 1) and measuring the performance of the forward prediction models in different scenarios. The resulting overall accuracy for evaluation tests in which the EC tokens were not randomised, randomised within the same class, and randomised across different classes was 49.6%, 41.3% and 38.3%, respectively. The benefit of using EC numbers becomes apparent upon grouping the test samples by class (Supplementary Table 8, Fig. 4a and Supplementary Figs. 6a and 7a) and linking each to their own sample size (Fig. 4b and Supplementary Figs. 4b–d, 6b–d and 7b–d). Tests belonging to EC-level 3 subclasses containing a large number of samples perform well even with incorrect EC numbers. The larger data set size of Oxidoreductase- (class 1) and transferase- (class 2) reactions, and for transferases also the homogeneity of the chemical space covered by substrates/products, make the presence of the EC number non-essential to determine the outcome of the chemical transformation. On the other hand, the accuracy among small and medium-sized classes drops, inversely correlating with the number of test and training samples for each EC-level 3 category over all classes (Supplementary Fig. 15). The performance on the ligases (class 6) shows a marginal increase from 32.3% to 33.9% when EC numbers are randomised within the same class and drops to 8.1% when EC numbers are randomised across different classes, suggesting that the attention relies on the class level of the EC number for accurate predictions. As a general trend over all classes and experiments, the accuracy increases while increasing the number of predictions to match (top-$k$, $k \in \{1, 2, 3, 4, 5\}$), with the biggest effect between $k = 1$ and $k = 2$.

Therefore, the models can use EC numbers to learn the biocatalysis signal as shown in Supplementary Fig. 8 for a selection of successfully predicted reactions. On the other hand, the analysis of incorrectly predicted reaction outcomes, reported in Fig. 5, highlights peculiar patterns. Reactions (1) and (2) are both catalysed by an oxidoreductase acting on the CH-NH$_2$ group

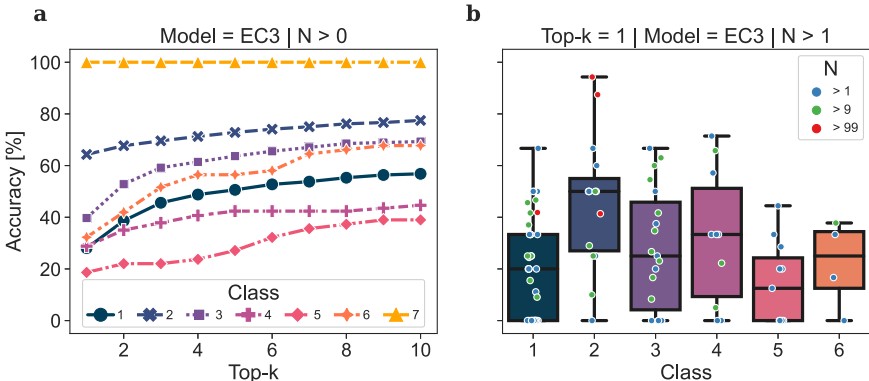

**Fig. 4 Class-wise accuracy for the forward model trained on EC3. a** The top-k prediction accuracy for each class show significant differences among classes caused by the number of available samples per EC-level 3 category. The accuracy of **b** top-1 predictions per EC-level 3 category. Each dot represents an EC-level 3 subclass coloured by the number of test samples N. Large EC-level 3 subclasses (red) greatly influence the performance of predicting transferase-catalysed reaction (class 2) outcomes. Oxidoreductase-catalysed reactions (class 1) are distributed among many EC-level 3 subclasses, causing a lower performance compared to other classes with fewer samples overall. Detailed accuracies for top-2 and top-5 predictions can be found in Supplementary Fig. 4.

**Fig. 5 Inspection of forward predictions labelled as incorrect.** For each reaction, the ground truth is shown in black while the prediction is shown in red. The reactions are catalysed by (1, 2) oxidoreductases acting on the CH-NH$_2$ group of donors with oxygen as acceptor, (3) a zeatin 9-aminocarboxyethyltransferase, (4) a cyclic-CMP phosphodiesterase, (5) a chloromuconate cycloisomerase, (6) and a pantothenate synthetase.

of donors. The predicted reaction (1) contains an excessive number of carbon atoms. The inferred product of the reaction (2) is equivalent to the ground truth, as the linear and cyclic forms are in equilibrium. (3) shows an example of the model correcting an error in the data set and predicting the correct stereochemistry. The discrepancies also highlight the possibility of false negatives due to the prediction of zwitterions. Concerning the diester hydrolase-catalysed reaction (4) and the intramolecular lyase reaction (5), it is worth noting that the data set contains few chemical reaction records with identical substrates and EC

numbers that result in different products. With sufficient data, the model would be able to recommend, given an EC-number, the various transformations of a single substrate into different products with corresponding confidence levels. However, due to the limited data volume and the random nature of the split into training, validation, and test sets, it is unlikely that all possible outcomes of a single substrate and EC number will be used for training. As a result, the two reactions are predicted incorrectly. (6) is an example of the model failing to predict the correct stereochemistry of a product. Prediction of correct

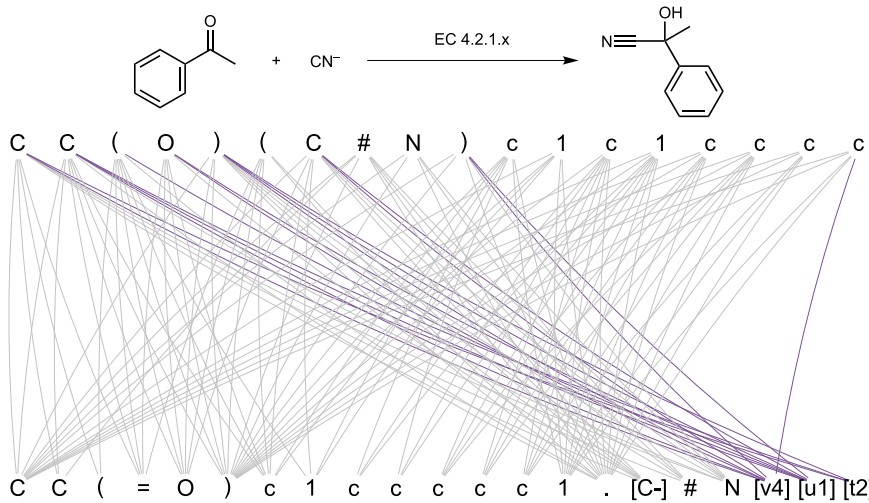

**Fig. 6 Analysis of the attention weights in the forward prediction models on reaction (6) from Supplementary Fig. 8.** The attention mapping between tokens representing EC numbers is highlighted in purple (reactant atom tokens are connected using grey curves). The curve thickness is proportional to the attention weight computed by the forward Molecular Transformer.

stereochemistry has been reported by Schwaller et al.[25] as a major challenge for the molecular transformer and is linked to the lack of coherent stereochemical information in the USPTO data set[33]. Similarly, the correct stereochemical prediction, affected by the limited data coverage on stereochemical examples, remains the major challenge for the model, especially when predicting reactions catalysed by isomerases (class 5, Fig. 4a). The removal of all stereochemical information from the predicted products increases the accuracy of isomerase-catalysed reaction prediction by a factor of two (Supplementary Fig. 5).

Inspired by the work of Schwaller et al.[18] we unboxed the forward prediction model to better understand how it exploits enzyme information. Figure 6 shows an example of an enzymatic reaction and the attention relationship between the EC and the product tokens. The EC tokens relate to the centre of the nucleophilic addition as well as the reactive nucleophile. A more extended analysis of the attention patterns and additional examples can be found in the supplementary information. The analysis of the attention weights confirms the capacity of the forward Molecular Transformer to use the EC token for discerning the enzymatic reaction centre while capturing enzymatic reaction rules.

**Backward prediction.** Here, we will use both the round-trip[26] and the top-$k$ accuracy for the evaluation of the backward model. Both metrics will provide evidence of how good the model is at proposing different enzymatic reactions that can lead to the desired target (round-trip accuracy) or the one specifically reported in the ground truth (top-$k$ accuracy). With a top-1 accuracy of 60%, the backward model has a behaviour similar to the forward model in the performance between and within classes (Fig. 7), as well as in the correlation between the size of the training samples and accuracy (Supplementary Fig. 16). In addition to the substrates, the model also predicts the enzyme EC-level 3 token; the accuracy of predicting EC numbers only is shown in Supplementary Fig. 10. The analysis of the model shows an exceptional performance on transferase-catalysed reactions (class 2), traceable to the two large EC-level 3 subclasses EC 2.3.1.x, and EC 2.7.8.x., which contain 17%, and 20% of all available samples, respectively (Fig. 7b). This analysis further explains the comparatively low prediction accuracy in the class of oxidoreductases (class 1) as it contains a large number of EC-level

3 subclasses, each small in size (Fig. 7 and Supplementary Table 4). Translocases are involved in catalysing the movement of molecules or ions across membranes. Together with the limited set of reaction records (191), this specific function causes the substrates and products to have lower diversity than those in other classes. The limited data reduces the statistical significance for this class, and we have thus opted to exclude Translocases from a detailed analysis.

The confusion matrix (Supplementary Fig. 12) provides further insight into the backward prediction performance. The model's ability to assign a product to the correct enzymatic class differs significantly between classes and is again influenced by the cohort of each class. Despite the larger population of the oxidoreductases, the split into many EC-level 3 subclasses causes the backward model to perform worse in predicting substrates for oxidoreductase-catalysed reactions than for hydrolase-, lyase- and ligase-catalysed reactions (Fig. 7a). However, the prediction of the enzyme class (EC number) only shows high accuracy (71.97%) for the class of oxidoreductases. This result shows that given a diverse reaction data set, the model can distinguish between classes but does not have enough data to predict the correct substrates. A challenge in terms of predicting the correct enzyme class are the isomerases (class 5), as they encompass intramolecular oxidoreductases, transferases and lyases. This is reflected in the relatively high misassignment of isomerases to oxidoreductases, transferases and lyases (classes 1, 2 and 4).

Similar to the forward reaction prediction model, we report a set of successful (Supplementary Fig. 14) and unsuccessfully predicted backward reactions (Fig. 8), together with their ground truth. The successful examples reflect the models' capability to predict substrates across enzyme classes. Among the unsuccessful ones, Example (1) highlights the prediction of a different EC-level 3 token to catalyse the reaction. Whereas the ground truth reaction is catalysed by an oxidoreductase acting on the CH-OH group of donors with oxygen as an acceptor (1.1.3.x), the prediction suggests the reaction to be catalysed by an oxidoreductase acting on the CH-OH group of donors with NAD$^+$ as an acceptor (1.1.1.x). This choice may reflect the respective number of training samples for the two classes (3,220 and 176 for 1.1.3.x and 1.1.1.x, respectively) and could be considered a viable alternative to the ground truth. Example (2) shows a correct EC-level 3 prediction (hexokinase). However, the substrate did not match the ground truth because the model predicted the acyclic

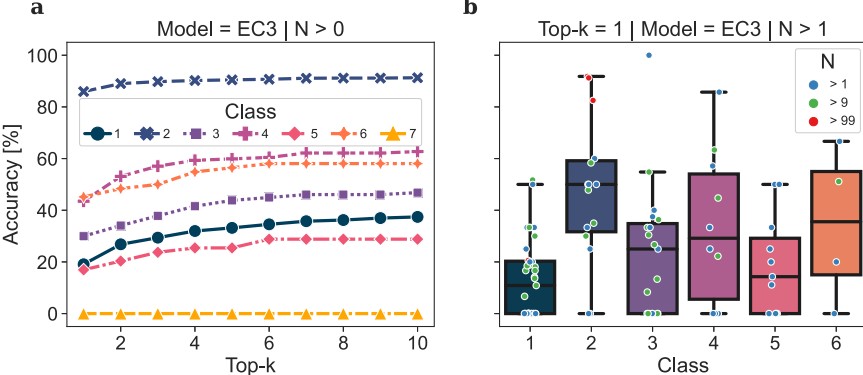

**Fig. 7 Class-wise accuracy for the backward model trained on EC3. a** The top-k prediction accuracies for each class (corresponding to EC-level 1) show significant differences among classes caused by the number of available samples per EC-level 3 category. The accuracy of **b** top-1 predictions per EC-level 3 category. Each dot represents an EC-level 3 category coloured by the number of test samples N. Large EC-level 3 subclasses (red) greatly influence the performance of predicting transferase-catalysed reaction (class 2) outcomes. Oxidoreductase-catalysed reactions (class 1) are distributed among many EC-level 3 subclasses, causing a lower performance compared to other classes with fewer samples overall. Detailed accuracies for top-2 and top-5 predictions can be found in Supplementary Fig. 9.

rather than the linear form of aldehydo-D-galactose. (3) is an example of the model adding stereochemistry information missing in the test data set. In the ground truth, only L-tyrosine is represented by an isomeric SMILES, while PAPS (3′-Phosphoadenosine-5′-phosphosulfate) and the product are represented in their racemic form. The model predicts both L-tyrosine and PAPS with the correct stereochemistry. In (4), the model predicts an alternative way to synthesise 2-fluorobenzoate. Rather than hydrolysing a coenzyme A thioester using a thioesterase, the model suggests an aldehyde dehydrogenase acting on the -CHO group of 2-fluorobenzaldehyde with $NAD^+$ as an acceptor. In contrast to the ground truth, the carboxylic acid can be obtained by mild oxidation of a commercially available substrate. Finally, for (5), the model fails to predict an enzymatic reaction for the synthesis of 3,5-dichloro-2-methylmuconolactone and falls back to a reaction learned from the USPTO data set.

**Retrosynthesis use-cases.** The trained forward and backward models allow us to extend the approach for template-free retrosynthesis prediction[26] to enzymatic reactions, introducing the first template-free biocatalysed synthesis planning tool (see the Methods section for details). Here, we present the predicted pathways for a selected number of target molecules and compare them to classical organic synthesis routes. We selected the target molecules from the RetroBioCat's curated set of biocatalysed pathways[34] based on the intersection of chemistry coverage in our data set ECREACT and the data set of RetroBioCat. In fact, the encoding of ECREACT and the RetroBioCat test set using rxnfp[35] shows that the RetroBioCat test set reactions are forming distinct clusters in the TMAP-embedded reaction space (Fig. 9a), in which the fraction of nearest neighbours from the set itself is consistently higher compared to reactions from ECREACT (Fig. 9b). This analysis highlights the different chemistry captured by the data sets and anticipates a poor performance for those RetroBioCat examples poorly covered in the ECREACT data set (see Supplementary Fig. 21 for a coverage-analysis of the reactions in the curated set from Finnigan[34] and ECREACT on a per-class basis).

In Fig. 10 we report the synthesis of the target molecules as recommended by the model using enzymatic transformations in mild conditions. Aminoalcohol (1) can be synthesised by regioselective transamination of the precursor dione, followed by reduction of the aminoketone with NADH as the hydride source.

This approach represents an alternative to gaseous hydrogen or other solid hydride sources typically employed in the reduction of carbonyls, which often represent a safety concern when employed already on gram-scale. Homoaspartate (2) can be accessed by a series of chemoselective enzymatic transformations of L-erythrose to the corresponding carboxylic acid, followed by regioselective dehydration to the α-ketoacid. Finally, the model infers that a transamination with glutamate on the newly-introduced keto functionality ensures the delivery of the target amino acid. Given the similar reactivity of the -OH groups within the substrate, such a series of transformations would require considerable effort using non-enzymatic approaches[36]. For the third example, the model predicts that 4-hydroxy-L-glutamic acid (3) can be obtained from oxidation of inexpensive L-hydroxyproline in the presence of $NAD^+$ (catalysed by EC 1.5.5.x), followed by a further oxidation of the aldehyde intermediate with EC 1.2.1.x and $NADP^+$. Enzymatic reactions enable oxidations to be also carried out in the presence of $O_2$, as exemplified by the prediction of the synthesis of α-ketoacid (4). The chemoselective oxidation of the amino group of L-tyrosine leaves the sensitive and electron-rich aromatic moiety unaltered and obviates the use of stronger oxidising agents. Lastly, the model predicts that an enzymatic Pictet-Spengler reaction catalysed by EC 4.2.1.x, can convert dopamine and the corresponding aldehyde to the alkaloid (S)-norlaudanosoline (5) enantioselectively, which typically requires the presence of organocatalysts or transition metals[37–39]. It is interesting to compare the routes suggested by our model with the ones from RetroBioCat[23]. In reaction (1) the starting substrate is styrene, which undergoes epoxidation in the presence of an epoxidase, followed by epoxide opening, partial oxidation of the primary alcohol to aldehyde and transamination. Homoaspartate (2) is instead shown to be obtained via aldol addition of sodium pyruvate on formaldehyde, followed by transamination with alanine. Similarly, RetroBioCat shows that 4-hydroxy-L-glutamic acid (3) can be prepared by treatment of pyruvic acid with glyoxylic acid in the presence of an aldolase, delivering the target compound after transamination with an amino donor. Pyruvic acid is also the substrate suggested for the synthesis of α-ketoacid (4), which is delivered upon reaction with phenol in the presence of a lyase. Alkaloid (S)-norlaudanosoline (5) is synthesised in a similar fashion as suggested by our model with a norcoclaurine synthase, which acts on the same primary amine and aldehyde substrates shown by the Molecular Transformer. With the

**Fig. 8 Inspection of backward predictions labelled as incorrect.** For each reaction, the ground truth is shown in black while the prediction is shown in red. The ground truth enzyme is marked with purple, the predicted enzyme with red. The model predicted (1, 4) different enzyme-catalysed reactions leading to the same product, (2) predicted a substrate with a different isomer, (3) corrected an erroneous data set entry and (5) was not able to predict an enzymatic reaction and fell back on a reaction learned from USPTO data.

exception of route (5), which is highly substrate-specific, one can appreciate the dissimilarity of the synthetic pathways suggested by our model when compared with RetroBioCat. While RetroBioCat algorithmically constructs a set of potential substrates using expert-curated rules, our model incorporates both the implicit reaction rules and its knowledge about the substrates contained within the training set into the generation of potential substrates. Based on the ECREACT training set, which contains mainly biosynthetic reactions, the suggested substrates resemble natural products with potentially high affinity towards the wild-type

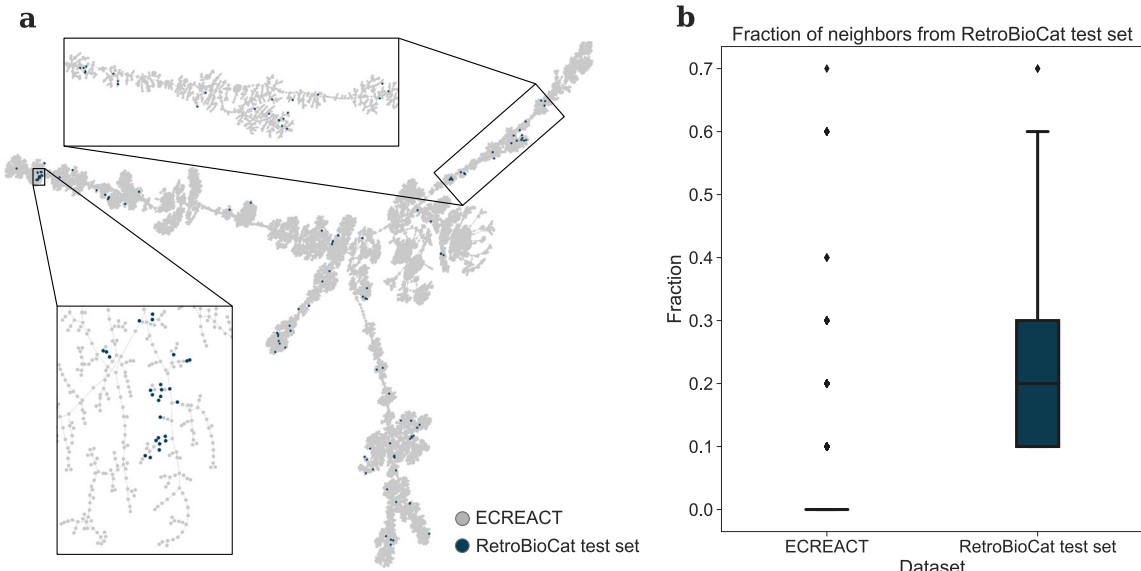

**Fig. 9 Distribution of rxnfp fingerprints for the reactions in the combined space of ECREACT (grey) and RetroBioCat test set reactions (blue), embedded with TMAP. a** The reactions from the RetroBioCat test set are forming distinct clusters in the combined reaction space. **b** For RetroBioCat test set (blue) reactions, the fraction of nearest neighbours ($k = 10$) from the set itself is consistently higher compared to reactions from ECREACT (grey).

**Fig. 10 Enzyme-catalysed synthesis of synthetically useful compounds under mild conditions.** (1) Aminoalcohol, (2) Homoaspartate, (3) 4-hydroxy-L-glutamic acid, (4) $\beta$-ketoacid and (5) (*S*)-norlaudanosoline.

enzymes. The use of commercial biocatalytic reaction-specialised data sets, not accessible to the authors and the general public, would likely lead to quite different and possibly more similar predictions to the ones suggested by RetroBioCat. However, even with the biosynthetic-specific nature of our data set, the model remains flexible when tasked to predict never-seen substrates by associating enzyme classes with the centre of the enzymatic reaction (Fig. 6 and Supplementary Fig. 18). As the molecular transformer-based model allows for retraining or fine-tuning using different data sets, these examples show that the substrate-scope can be narrowed down or expanded, depending on the use-case and available data. These properties open a way to synthetically useful compounds from a variety of different inexpensive substrates once experimentally validated.

## Discussion

We presented forward and backward prediction models based on Molecular Transformer trained on enzyme-catalysed reactions extended with EC (enzyme commission) numbers. Our results show that the Molecular Transformer performs well in predicting products given EC number and substrates, predicting substrates and EC number or EC number alone given a product. The enzymatic models reach an overall top-1 accuracy of 49.6%, 60% and 39.6% in forward, backward and round-trip accuracy, respectively. The accuracies correlate heavily with the amount of training data in each token class, presenting a major challenge given the limited data availability. In addition to the trained models for biocatalysed synthesis planning, we introduced an aggregated data set, ECREACT, containing preprocessed enzyme-catalysed reactions sourced from different publicly available databases. The primary limitation of the data set, and thus of the model, is the scarcity of samples for some classes, such as iso-merases, which results in an imbalanced data set. However, as interest in biocatalysis grows and the research community embraces open data, we anticipate an increase in the quantity and quality of available training data, as well as experimental valida-tion of proposed synthetic routes. In addition, the data has been sourced from databases containing mainly biosynthesic reactions due to the lack of publicly available data on the biocatalysed synthesis of non-natural products, leading the model preferably suggest natural products as substrates. Despite this bias, we showed that the model generalises by learning the association between enzyme classes and the centres of the enzymatic reac-tions, which is a valuable property given the limited availability of biocatalytic reaction data. The research community will be able to build on the legacy of the present work to retrain models with higher accuracy and broader scope without the limitation of humanly curating reaction rules. Finally, we presented several use-cases based on well-understood pathways that showed how template-free machine learning models trained on enzymatic reactions can play an essential role in promoting the adoption of greener chemistry strategies in daily laboratory work.

## Methods

**Transfer learning**. We used the USPTO data set, which contains 1 million organic chemical reactions, together with the more specific enzymatic reaction data set to train the molecular transformer using multi-task transfer learning. The USPTO data is used to learn general chemistry knowledge and the SMILES grammar, as the comparatively small enzymatic reaction data set does not provide sufficient information for these tasks. This approach was previously successfully applied to carbohydrate reactions by Pesciullesi et al.[33]. The reactions in the USPTO data set are encoded as so-called reaction SMILES, using the same convention of Schwaller et al.[25]. An example is the reaction SMILES CC(=O)O.OCC>OS(=O)(=O) O>CC(=O)OCC.O encoding a Fischer esterification. The main conceptual dif-ference lies in extending the reaction SMILES tokeniser to handle enzymatic reactions represented by the EC number as detailed in the section Preprocessing. The Molecular Transformer models were implemented following the protocol introduced by Schwaller et al.[25]. Multi-task transfer learning was implemented, as

described by Pesciullesi et al.[33], using a convex weighting scheme for USPTO and ECREACT, 9 and 1, respectively. Both encoder and decoder were of type *trans-former* with 6 layers, word vectors and RNN of size 512, the gradient was accu-mulated 8 times with a maximum vector norm of 0.0, and *adam* was used as an optimiser ($\beta_1 = 0.9$, $\beta_2 = 0.998$). Batch size was set to 4096, and the batch type as well as the gradient normalisation method to *tokens*. The learning rate was set to 2.0 with *noam* as decay method. Dropout and label smoothing ($\epsilon$) were set to 0.1. Parameter initialisation was disabled and position encoding enabled. All models were trained using a version of OpenNMT[40] adapted for the Molecular Transformer[41].

**Preprocessing**. The standard definition of a reaction SMILES was extended to include EC numbers (e.g. the reaction catalysed by the maltose alpha-D-glucosyltransferase is written as A|5.4.99>>B, where the SMILES for D-maltose and α,α-trehalose have been replaced by A and B for brevity). We denote this extension to reaction SMILES enzymatic reaction SMILES.

We adapted the tokenisation operation used by Schwaller et al.[25] for the molecular transformer to handle enzymatic reaction SMILES. EC-levels 1-3 are treated as unique tokens to enable the transformer to learn the hierarchical structure of the EC numbering scheme. Because digits are already used to represent ring closures in SMILES, a number prefix is added to each level (v for EC-level 1, u for EC-level 2 and t for EC-level 3) during tokenisation. In addition, each EC token is encapsulated in brackets to simplify the tokenisation and detokenisation process. An example tokenisation of an enzymatic reaction SMILES is shown in Supplementary Fig. 22.

Finally, the resulting tokenised data set was split into a training, validation and test set (90%, 5% and 5%, respectively). The training set was sampled so that none of the products contained within it are present in the training and the validation data sets.

**Retrosynthesis routes prediction**. We adapted the methodology proposed by Schwaller et al.[26], extending the retrosynthetic routes' prediction to handle enzyme information using the EC number format. The hyper-graph exploration algorithm, at each step, proposes disconnections using the backward model and computes a score for each prediction, in a Bayesian sense, based on the confidence of the forward model reweighted by the SCScore[42] measured on the precursors. The pathways are then prioritised, exploiting the score by using beam search until a terminating condition is satisfied, i.e., commercial availability of the precursors (see Supplementary Fig. 23).

For the analysis of the targets from[23] we used an interactive version of the approach, where the backward Molecular Transformer allowed us to explore the synthetic routes iteratively until reaching commercially available precursors and proposing, at the same time, enzymes (represented up to EC level 3) that catalyse the corresponding reaction.

We based the selection of the targets on a comparative analysis of the coverage of the chemistry embedded in the reactions from the RetroBioCat[23] test set and the ECREACT data set. We annotated reaction SMILES for each step of the biocatalytic cascades considered in the test set from Finnigan et al.[23], excluding solvents information. For each reaction SMILES we extracted fingerprints using rxnfp[35] and we computed among the k-nearest neighbours ($k = 10$), the fraction of neighbours belonging to RetroBioCat test set. The visualisation of the embedded reactions was generated using TMAP[31].

## Data availability
The ECREACT data set is publicly available at the URL https://github.com/rxn4chemistry/biocatalysis-model. The trained models are made publicly available as part of IBM RXN for Chemistry (https://rxn.res.ibm.com/).

## Code availability
Code is available at the URL https://github.com/rxn4chemistry/biocatalysis-model.

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

## Acknowledgements

This publication was created as part of NCCR Catalysis (grant number 180544), a National Centre of Competence in Research funded by the Swiss National Science Foundation.

## Author contributions

The project was conceptualised by D.P., M.M., F.P. and T.L.; D.P. and M.M. curated the data, developed the methodology, and carried out the formal analysis, investigation and validation. D.P., M.M. and Y.T. created the visualisations and software. D.P., M.M., Y.T. and A.C. wrote the original draft. A.C. validated the test set and curated the data for the use cases. T.L. supervised and administered the project, reviewed and edited the writing and acquired funding and resources.

## Competing interests

The authors declare no competing interests.
