## [Peer Review File · Nature Communications]

REVIEWER COMMENTS

Reviewer #1 (Remarks to the Author):

The manuscript by Probst et al describes the use of a Molecular Transformer to predict products of biocatalysed reactions. To that end, the authors employ a new token scheme based on the EC classification and show its importance for the model performance. My comments are based not only on the machine learning merit of the study but also from a medicinal and organic chemistry vantage point. While the machine-learning model itself is not new (this is clearly stated by the authors) and I am not overly impressed with the top-k accuracy values, I do believe the highlight of the work is encoding enzymes through their EC characters. This can create a precedent that might be further expanded to other classification systems, such as the Anatomical Therapeutic Chemical code, and different research questions. The work is very well written but, at times, a bit too overwhelming for a more general chemistry audience, in particular experimentalists. The manuscript is certainly publishable, but some clarifications are warranted. My concerns/comments are listed below by order of appearance in the manuscript.

- 1) Why did the authors choose a model trained on the USPTO dataset as baseline? How is it meaningful for the research question in this study?
- 2) "The forward prediction model achieves an accuracy of 49.6% and 62.7%, top-1 and top-5 respectively, while the single-step retrosynthetic model shows a round-trip accuracy of 39.6% and 42.6%, top 1 and top-10, respectively" For consistency, I would suggest presenting the top-1/5 or top1/10 in both cases.
- 3) "An extensive analysis of the data set pinpoints the performance to the enzyme class distribution of the training data, with the forward prediction model ranging from an accuracy of 18.6% and round-trip accuracy of 1.7% for isomerases, the most scarce class, to a forward prediction accuracy of 64.4% and a round-trip prediction accuracy of 60.5% for transferases, the most abundant class." The sentence is a bit convoluted. I would suggest rephrasing or dividing it in two.
- 4) The manuscript would benefit from Figure 1 demonstrating the data distribution across different classes. I think Figure 10 is overwhelming. Maybe the middle pie chart is enough for the main manuscript. The caption would then link to the full Figure 10 (moved to SI) and Tables S3-4 for details.
- 5) I think that top-k accuracy is not the best way of assessing models, but in all honesty I am not able to suggest a viable and validated alternative that is simultaneously convincing for a cheminformatics and medicinal chemist/chemical biologist audience. There are however caveats to top-k metrics and maybe the authors could discuss them a bit in the manuscript (a few sentences would be enough) for better context and targeted at a more general audience.
- 6) The authors point towards a limitation on the study. The database does not seem to be homogeneous in several aspects that are key for a correct comparison between predictions and ground truth. Although this does not invalidate the conclusions, I would suggest introducing a word of caution or (ideally) a section regarding limitations in the dataset. This would probably mean de-emphasizing the top-k values discussion in the manuscript to a certain extent, since those limitations are likely skewing the performance evaluation. If they are not skewing, which data curation measures were taken? How many database entries have issues in the stereochemistry (unassigned, non-conclusive, wrong) protonation states and/or others (either on substrate or product sides)? I reckon this is an important point so that each reader can make an informed and independent assessment on the value of the database.
- 7) Figure 3: rather than incorrect predictions, the figure should be captioned with something else because most "incorrect" predictions are indeed deficiencies in the source data, e.g. the ground truth for reaction 3 (Figure 3) has two errors.
- 8) I do not quite follow the justification for reaction 5. Maybe depict the reaction the authors allude to.
- 9) Typo in page 11, line 14: "int"
- 10) "With the exception of route (5), which is highly substrate-specific, one can appreciate the dissimilarity of the synthetic pathways suggested by our model when compared with RetroBioCat, which opens the way to synthetically useful compounds from a variety of different inexpensive substrates." This reads a bit too assertive without experimental evaluation for proof-of-concept. I would suggest either rephrasing or adding an additional sentence saying these are still predictions and would warrant experimental work to more accurately assess the value of the recommended pathways.

Overall, this is a nice piece of work and would recommend publication if these and other comments are adequately clarified.

I chose not to remain anonymous: Tiago Rodrigues

Reviewer #2 (Remarks to the Author):

The authors present a transformer model trained using transfer learning first on the USPTO dataset, and then on a bespoke 'ECReact' dataset which the authors constructed. This is very similar to recent work (Chem. Sci., 2021, 12, 8648), which the authors do reference. The main difference being the use of EC number rather than enzyme name as a reaction token, and the construction of an alternative enzyme dataset. However, these differences do not seem to have translated into an improvement in accuracy over the previous work. The authors do present a working retrosynthesis tool using their model, which goes beyond what was presented in Chem. Sci., 2021, 12, 8648. Comparison of the results of this retrosynthesis tool to suggestions by the retrobiocat tool are also interesting.

I believe this work will be of interest to others working on computer-aided synthesis planning. However, the work presented does not seem a particularly big step beyond previously published work (Chem. Sci., 2021, 12, 8648), and I wonder whether the accuracies presented mean this is unlikely to be taken up by practitioners of biocatalysis or metabolic engineering?

There are a number of points which I think need to be addressed, detailed below:

Introduction

I would expect a few more citations in the opening paragraph of the introduction to support some of the statements made there.

Some sentences the authors might want to revisit:

Page 2 line 19 – "enzymatic reactions to catalyse organic synthesis". Small thing, but it's the enzyme (not the reaction) that is the catalyst. Just needs re-wording.

Page 3 line 10 – "One of the first work" – the sentence reads strangely and might want to be re-written.

Page 4 line 19-20 – "The use of a backward model allows to predict substrates and catalysing enzyme classes given a target product." - This also reads badly

Page 4 line 17 onwards – Large parts of this feel like it should be in the results. This paragraph seems to go beyond introducing the aims and objectives of the work and into describing specifically what was done. I would have thought information about constructing the ECREACT dataset should be in the results rather than the introduction?

Results and Discussion

Page 5 line 15 – The EC3 data is presented without any information about what this is or where it's come from. In the materials and methods this is presented. Maybe it would be useful to start the results section with a brief description of the construction of this dataset (rather than talking about this in the introduction), and specifically what the EC3 data is. This seems an important part of the narrative before we get into the forward or reverse prediction?

It does seem that a number of results and discussion are currently presented in the materials and methods section, which doesn't seem correct. I would have thought the construction and analysis of the ECReact dataset, and the analysis around the different EC levels belongs in the results and discussion section. Furthermore, figures 10 and 11 are clearly an analysis of the dataset rather than a method.

Page 11 line 5 – Figure 12 Shouldn't this figure simply be the next numbered figure in the results section?

(so figure 12 should actually be figure 5?)

Figure 8 – are the retrobiocat reactions similar to each other simply because they are in cascades, so the reaction participants are similar? Taking metabolic pathways from BRENDA or Rhea, would a similar result be achieved compared the entire ECRreact dataset?

I think the point raised about the differences between the routes suggested by RetroBioCat and the transformer presented here is an interesting one. Is there a tendency of the model presented to suggest routes which might be more closely aligned with metabolism, given that all the training data is for reactions from metabolism?

How similar are the reactions suggested by the retrosynthesis tool to reactions in the dataset? Are these reactions in the dataset?

Do the reactions suggested by the tool have an advantage over the retrobiocat suggestions in that they are likely to be similar to natural substrates, with a reasonable likelihood of being accepted (as opposed to the more general rules used by retrobiocat)? However possibly the scope for what will be suggested is smaller.

Materials and methods

As mentioned, there are results and discussion presented here which should be moved to provide a better stand-alone narrative in the actual results and discussion section

General questions and comments

Are the accuracies presented too low to be of much practical use? Perhaps this could change with a larger dataset? How much data do the authors envisage needing to achieve reliable predictions, and is this feasible in the medium-near future? This seems a major part of the conclusions presented by the authors. How rapidly is the available data increasing year on year?

Biocatalysis vs Biosynthesis

There is a focus on biocatalysis in this work, when perhaps biosynthesis should be discussed given that the dataset comes from a number of databases detailing reactions in metabolism. A number of points on this theme are detailed below:

Is there scope to use the retrosynthesis tool for metabolic engineering?

I imagine the contents of the dataset constructed in this work vs Chem. Sci., 2021, 12, 8648 is quite different, as this works takes reactions from BRENDA, Rhea, PathBank and MetaNetX, whilst the referenced work takes reactions from Reaxys. The reaxys dataset is likely more suitable for making suggestions for the use of enzymes for organic synthesis (biocatalysis), whilst the dataset constructed by the authors is likely more suited to biosynthesis or metabolic engineering? This difference in the datasets could be better explored in the manuscript?

Following this point, how well can the model be expected to generalise to synthetic substrates if the dataset is limited to only the natural substrates of enzymes, even if/as this dataset grows.

What is the advantage of the approach described by the authors over that adopted by RetroRules, which utilises a similar dataset. (<https://retrorules.org/>) (<https://doi.org/10.1093/nar/gky940>). Perhaps a comparison to RetroPath would be worth including?

Reviewer #3 (Remarks to the Author):

The manuscript by Probst and colleagues presents a biosynthesis planning system based on machine learning algorithms. The work is a follow up of previous publications from the group, which has developed a

system called Molecular Transformer. Here, the authors generalise the use of the program by adopting a tokenisation system based on EC classes as well as an extension of the retrosynthetic algorithm.

The original Molecular Transformer was based on a rich dataset from the USPTO. In order to associate the reaction SMILES with the EC numbers, the authors develop a new dataset that extracts such information from common biochemical databases like MetaNetX. In that way, text information describing the reaction can be incorporated as well. However, EC numbers might not be available when considering reactions like those found in the USPTO data set, which method are proposing the authors in order to process such data set? The answer is not completely clear in the manuscript.

The main purpose of this study seems to be a comparison with the RetroBioCat retrosynthesis tool as a benchmark, although it is not clear if the proposed tool outperforms RetroBioCat.

One issue is that even if accuracies seem to be higher than in the original proposed Molecular Transformer, they are still too low for cases that are well known to be hard to predict, such as for isomerases (EC 5). Therefore, there is no substantial progress from the new tool in that direction.

The writing is at some points awkward, for instance trivial sentences like "enzyme-catalysed reactions are usually performed in water..." or "the reaction classes' statistics of Finnigan". Moreover, the Github link is not working.

We would like to express our gratitude to the referees for carefully reading our manuscript and providing constructive feedback that significantly improved the paper's quality. We considered all the points raised. The referee's suggestions regarding the text and grammar have been incorporated into the revised version. In the following, we report a detailed response for each of the feedback: the referee's comments are repeated (*enumerated and written in italic*), followed by a detailed response by the authors prefixed with "Answer:" and the amended text in the article when that was needed.

Reviewer 1

1. *My comments are based not only on the machine learning merit of the study but also from a medicinal and organic chemistry vantage point. While the machine-learning model itself is not new (this is clearly stated by the authors)*

Answer: We would like to thank referee #1 for the positive feedback. The architecture of the model was initially published in 2019 by the IBM team and it has been used before for the prediction of traditional organic chemistry reactions (see the several group's publications) and, more recently, for forward enzymatic reaction prediction (Kreutter et al., 2021). However, the training of a model for backwards / retro predictions is a completely new use of this architecture in the context of enzymatic reaction. Moreover, unlike the work of Kreutter et al., 2021, we are encoding enzymatic reaction data using the EC number to make a better use of all those chemical reaction records that have little statistical relevance. We modified the abstract and the introduction to convey this message more clearly

2. *The work is very well written but, at times, a bit too overwhelming for a more general chemistry audience, in particular experimentalists.*

Answer: We tried to strike a balance between describing the work for a general chemistry experience and providing sufficient technical details to support reproducibility by other groups/researchers. Moreover, we made the entire work more easily consumable by giving access to the deployed model on a graphical user interface to lower the adoption barrier for non-ML experts in their daily research work. Some of the next points have been carefully considered to make the work less overwhelming and more easily readable by a larger audience.

3. *Why did the authors choose a model trained on the USPTO dataset as baseline? How is it meaningful for the research question in this study?*

Answer: The volume of enzymatic reaction data alone is insufficient (70K entries) to ensure proper SMILES grammar learning. The larger USPTO data set is required in addition to the enzymatic reaction data for the model to learn the SMILES grammar using multi-task transfer learning. This approach has been already validated by Pesciullesi et al. (<https://www.nature.com/articles/s41467-020-18671-7>), who demonstrated the use of transfer learning when training predictive models with small data sets.

Added: "The USPTO data is used to learn general chemistry knowledge and the SMILES grammar, as the comparatively small enzymatic reaction data set does not provide sufficient information for these tasks."

Page 20, lines 5-7

4. *"The forward prediction model achieves an accuracy of 49.6% and 62.7%, top-1 and top-5 respectively, while the single-step retrosynthetic model shows a round-trip accuracy of 39.6% and 42.6%, top 1 and top-10, respectively" For consistency, I would suggest presenting the top-1/5 or top1/10 in both cases.*

Changed to: "The forward prediction model achieves an accuracy of 49.6%, 63.5%, and 68.8% top-1 and top-5, and top-10 respectively, while the single-step retrosynthetic model shows a round-trip accuracy of 39.6%, 42.3%, and 42.6%, top-1, top-5, and top-10, respectively."

Page 5, lines 1-3

5. *"An extensive analysis of the data set pinpoints the performance to the enzyme class distribution of the training data, with the forward prediction model ranging from an accuracy of 18.6% and round-trip accuracy of 1.7% for isomerases, the most scarce class, to a forward prediction accuracy of 64.4% and a round-trip prediction accuracy of 60.5% for transferases, the most abundant class." The sentence is a bit convoluted. I would suggest rephrasing or dividing it in two.*

Answer: We have removed this sentence from the introduction based on comments by reviewer 2. Their reasoning, which we agree with, is that it belongs in the results section.

6. *The manuscript would benefit from Figure 1 demonstrating the data distribution across different classes. I think Figure 10 is overwhelming. Maybe the middle pie chart is enough for the main manuscript. The caption would then link to the full Figure 10 (moved to SI) and Tables S3-4 for details.*

Answer: We believe that the reference to Figure 10 was referring to Figure 2. We transferred the entire figure to the SI (now Figure S3) while keeping only the middle section in the main manuscript as Figure 2.

7. *I think that top-k accuracy is not the best way of assessing models, but in all honesty I am not able to suggest a viable and validated alternative that is simultaneously convincing for a cheminformatics and medicinal chemist/chemical biologist audience. There are however caveats to top-k metrics and maybe the authors could discuss them a bit in the manuscript (a few sentences would be enough) for better context and targeted at a more general audience.*

Answer: We don't fully understand the general comment about the top-k metric's caveats. We can confirm that, while not perfect, top-k accuracy emerged as the most commonly used metric to evaluate comparable tasks (e.g.

<https://pubs.rsc.org/en/content/articlelanding/2021/sc/d1sc02362d>,

<https://pubs.rsc.org/en/content/articlehtml/2020/sc/c9sc03666k>).

8. *The authors point towards a limitation on the study. The database does not seem to be homogeneous in several aspects that are key for a correct comparison between predictions and ground truth. Although this does not invalidate the conclusions, I would suggest introducing a word of caution or (ideally) a section regarding limitations in the dataset. This would probably mean de-emphasizing the top-k values discussion in the manuscript to a certain extent, since those limitations are likely skewing the performance evaluation. If they are not skewing, which data curation measures were taken? How many database entries have issues in the stereochemistry (unassigned, non-conclusive, wrong) protonation states and/or others (either on substrate or product sides)? I reckon this is an important point so that each reader can make an informed and independent assessment on the value of the database.*

Answer: The limitations of the data set do indeed skew the performance evaluation. We identified the uneven distribution of training samples across classes and sub-classes as the primary issue with the data set and addressed it through a split analysis (see Figures 5 and 8, as well as Figures

S9 and S10). The questions about stereochemical issues, protonation states, and so on can only be addressed by manual curation strategies that are human intensive and cannot be dealt with by any existing automatism. We used the available dataset as ground truth because data curation is beyond the scope of this work. In addition, we made the entire dataset available to others so that they could use and analyse it more thoroughly and perhaps even improving its quality in the future.

Having said that, it is worth noting that we were able to demonstrate that both the forward and backwards models are resilient to errors in the ground truth (Figures 7 and 11, respectively). We added the following text to the conclusion to highlight the data set's limitations.

Added: "The primary limitation of the data set, and thus of the model, is the scarcity of samples for some classes, which results in an imbalanced data set. However, as interest in biocatalysis grows and the scientific community embraces open data rules, we predict an increase in the quantity and quality of available training data, as well as experimental validation of proposed synthetic routes. "

Page 18, lines 11-16

9. *Figure 3: rather than incorrect predictions, the figure should be captioned with something else because most "incorrect" predictions are indeed deficiencies in the source data, e.g. the ground truth for reaction 3 (Figure 3) has two errors.*

Answer: This is indeed true, and we changed the captions in Figures 7 and 11 accordingly. We chose a more neutral caption as shown below.

Changed to: "Inspection of forward predictions labelled as incorrect." and "Inspection of backward predictions labelled as incorrect."

Page 8, line 1 and page 13, line 1

10. *I do not quite follow the justification for reaction 5. Maybe depict the reaction the authors allude to.*

Answer: We hope that the updates shown below make the issue clear. The reaction is, in fact, depicted as the ground truth, and the correct reaction / product (according to the test set) in black (substrate + enzyme) and red (product).

X | 1.2.3.4 >> Y
X | 1.2.3.6 >> Z

Discarding EC-level 4 of the two reactions shown above leads to:

X | 1.2.3 >> Y (this reaction was part of the training set)
X | 1.2.3 >> Z (this reaction was part of the test set)

Changed to: "Concerning the diester hydrolase-catalyzed reaction (4) and the intramolecular lyase reaction (5), it is worth noting that the data set contains few chemical reaction records with identical substrates and EC numbers that result in different products. With sufficient data, the model would be able to recommend, given an EC-number, the various transformations of a single substrate into different products with corresponding confidence levels. However, due to the limited data volume and the random nature of the split into training, validation, and test sets, it is unlikely that all possible outcomes of a single substrate and EC number will be used for training. As a result, the two reactions are predicted incorrectly."

Page 6, lines 18-22 and page 7, lines 1-4

11. *Typo in page 11, line 14: "int".*

Changed to: "low prediction accuracy in the class of oxidoreductases"

Page 10, line 13

12. *With the exception of route (5), which is highly substrate-specific, one can appreciate the dissimilarity of the synthetic pathways suggested by our model when compared with RetroBioCat, which opens the way to synthetically useful compounds from a variety of different inexpensive substrates." This reads a bit too assertive without experimental evaluation for proof-of-concept. I would suggest either rephrasing or adding an additional sentence saying these are still predictions and would warrant experimental work to more accurately assess the value of the recommended pathways.*

This is true. We changed the sentence as shown below to make it less assertive.

Changed to: "which opens a way to synthetically useful compounds from a variety of different inexpensive substrates once experimentally validated."

Page 16, line 11 and Page 17, lines 1-2

Reviewer 2

1. *The main difference being the use of EC number rather than enzyme name as a reaction token, and the construction of an alternative enzyme dataset.*

Answer: We would like to thank reviewer 2 for the constructive feedback. We would like to stress that the construction of an alternative enzyme dataset is a by-product and not the main scope of this work. Beyond using EC numbers as reaction token to take advantage of those reaction records with little statistical significance, the main novelty of the current work is the use of the molecular transformer for backward prediction to construct retrosynthetic pathways. We adapted the abstract to reflect this fact better.

2. *However, the work presented does not seem a particularly big step beyond previously published work (Chem. Sci., 2021, 12, 8648), and I wonder whether the accuracies presented mean this is unlikely to be taken up by practitioners of biocatalysis or metabolic engineering?*

Answer: We stress again that prior art using Molecular Transformer was only capable of predicting forward reaction outcomes and could not be used to construct retrosynthetic pathways. We adapted the abstract and introduction to make the contribution to the field of biocatalysis clearer.

3. *I would expect a few more citations in the opening paragraph of the introduction to support some of the statements made there.*

Answer: We added the following additional citations to support the statements in the opening paragraph:

- Trewavas, A. Malthus Foiled Again and Again. *Nature* **2002**, *418* (6898), 668–670. <https://doi.org/10.1038/nature01013>.
- Matlin, S. A.; Abegaz, B. M. Chemistry for Development. In *The Chemical Element: Chemistry's Contribution to Our Global Future*; García-Martínez, J., Serrano-Torregrosa, E., Eds.; Wiley-VCH, **2011**; pp 1–70.
- Duigou, T.; du Lac, M.; Carbonell, P.; Faulon, J.-L. RetroRules: A Database of Reaction Rules for Engineering Biology. *Nucleic Acids Research* **2018**, *47* (D1), D1229–D1235. <https://doi.org/10.1093/nar/gky940>.
- Mazurenko, S.; Prokop, Z.; Damborsky, J. Machine Learning in Enzyme Engineering. *ACS Catalysis* **2020**, *10* (2), 1210–1223. <https://doi.org/10.1021/acscatal.9b04321>.

Page 2, lines 2-9

4. *Page 2 line 19 – “enzymatic reactions to catalyse organic synthesis”. Small thing, but it’s the enzyme (not the reaction) that is the catalyst. Just needs re-wording.*

Changed to: "Although the ability to use enzymes as catalysts in the organic synthesis of chemical compounds gained widespread attention for large scale production"

Page 2, lines 19-21

5. *Page 3 line 10 – “One of the first work” – the sentence reads strangely and might want to be re-written.*

Changed to: "Currently, only rule-based methods for predicting biosynthesis pathways have been examined, such as the ATLAS of Biochemistry or RetroRules."

Page 3, lines 10-12

6. *Page 4 line 19-20 – “The use of a backward model allows to predict substrates and catalysing enzyme classes given a target product.” - This also reads badly*

Answer: We changed the text, and based on your previous comments, also added emphasis on the differentiation to previous work to clarify the novelty of our data-driven backward model.

Changed to: "Compared to the previous work by Kreutter et al., a backward model allows to predict substrates and enzyme classes given a target product, enabling retrosynthetic pathway prediction. In addition, we incorporate the EC (enzyme commission) number into the reaction SMILES, rather than encoding enzymes with their natural language name."

Page 4, lines 20-24

7. *Page 4 line 17 onwards – Large parts of this feel like it should be in the results. This paragraph seems to go beyond introducing the aims and objectives of the work and into describing specifically what was done. I would have thought information about constructing the ECREACT dataset should be in the results rather than the introduction?*

Answer: As this is already being discussed extensively in the Results and Discussion and Methods sections (see further answers), we have shortened the discussion on the data set in the introduction.

Removed: “The resulting data set contains more than 62,000 unique enzymatic reactions.”

Removed: “The USPTO data set contains 1 million reactions without enzymatic information. These reactions acted as a training set for learning the general knowledge of chemical reactions and the SMILES grammar.”

Removed: “An extensive analysis of the data set establishes a connection between performance and the enzyme class distribution of the training data. For the most scarce class, the isomerases, the forward prediction accuracy is 18.6% and the round-trip accuracy 1.7%, while for the most abundant class, the transferases, the forward prediction accuracy is 64.4% and the round-trip accuracy 60.5%.”

Removed: “The forward and backward (substrate + EC -> product and product -> substrate + EC, respectively) models were then trained using multitask transfer learning on the ECREACT and the USPTO data sets (see Data Sets and Model Training).”

8. *Page 5 line 15 – The EC3 data is presented without any information about what this is or where it's come from. In the materials and methods this is presented. Maybe it would be useful to start the results section with a brief description of the construction of this dataset (rather than talking about this in the introduction), and specifically what the EC3 data is. This seems an important part of the narrative before we get into the forward or reverse prediction?*

Answer: While we think the data set will provide useful to the field of machine learning for biocatalysis, it is not the main result of the study. To improve readability and wider audience interest, we prefer to refer the reader to the Methods section. We changed the text with a reference to the appropriate section for the reader to learn more about the data set.

Changed to: "The Dataset was constructed following the details reported in the Methods, Data Sets and Model Training. We split the EC3 data set (n=56579) into a test and a training set, enforcing a zero overlap between the product distributions of the two ensembles, i.e. no product molecule present in the test set appears in the training set."

Page 5, lines 6-9

9. *It does seem that a number of results and discussion are currently presented in the materials and methods section, which doesn't seem correct. I would have thought the construction and analysis of the ECRReact dataset, and the analysis around the different EC levels belongs in the results and discussion section. Furthermore, figures 10 and 11 are clearly an analysis of the dataset rather than a method.*

Answer: We do not consider the dataset the main outcome of this work, but rather an important byproduct that will help stimulate the use of machine learning in biocatalysis.

10. *Page 11 line 5 – Figure 12 Shouldn't this figure simply be the next numbered figure in the results section? (so figure 12 should actually be figure 5?)*

Answer: Indeed. As the sentence becomes superfluous with the changes, we removed it entirely.

Removed: Figure 12 shows the backward model performance (round-trip and top-k accuracy) for the EC-level 3.

11. *Figure 8 – are the retrobiocat reactions similar to each other simply because they are in cascades, so the reaction participants are similar? Taking metabolic pathways from BRENDA or Rhea, would a similar result be achieved compared the entire ECRReact dataset?*

Answer: The reactions from RetroBioCat are clustered together because they have similar reaction mechanisms and do not show cascades of metabolic pathways.

12. *I think the point raised about the differences between the routes suggested by RetroBioCat and the transformer presented here is an interesting one. Is there a tendency of the model presented to suggest routes which might be more closely aligned with metabolism, given that all the training data is for reactions from metabolism?*

Answer: Naturally, a data-driven model inherits biases present in the training data set. ECREACT's sources are skewed towards metabolic reactions and this is reflected in the suggestions of our trained models.

13. *How similar are the reactions suggested by the retrosynthesis tool to reactions in the dataset? Are these reactions in the dataset?*

Answer: They are similar to ones in the training set (ML) as shown in Figure 8.

14. *Do the reactions suggested by the tool have an advantage over the retrobiocat suggestions in that they are likely to be similar to natural substrates, with a reasonable likelihood of being accepted (as opposed to the more general rules used by retrobiocat)? However possibly the scope for what will be suggested is smaller.*

Answer: When comparing to RetroBioCat, we encountered the challenge of not having access to all of the data used to build RetroBioCat. RetroBioCat, to our surprise, only shares a small set of data, not the entire set used to construct the chemical reaction rules. This precludes more thorough analysis and comparison. Our model's recommendations reflect the training data, which was largely derived from reactions containing natural substrates. However, the Molecular Transformer can learn the chemical transformation rules (for a proof, see <https://www.science.org/doi/10.1126/sciadv.abe4166>), and we show in this paper (see Figure 12 and section "Attention Analysis") that the model can learn the importance of the EC token in relation to the reaction centres. As a result, the trained model can successfully extend the learned biochemical transformation rules to chemical structures other than natural products.

15. *As mentioned, there are results and discussion presented here which should be moved to provide a better stand-alone narrative in the actual results and discussion section*

Answer: See answer to comment 14.

16. *Are the accuracies presented too low to be of much practical use? Perhaps this could change with a larger dataset? How much data do the authors envisage needing to achieve reliable predictions, and is this feasible in the medium-near future? This seems a major part of the conclusions presented by the authors. How rapidly is the available data increasing year on year?*

Answer: Our previous publication on (traditional organic chemistry) backward / retro reaction prediction has gained a lot of attention by the entire chemistry community (<https://pubs.acs.org/doi/full/10.1021/acscentsci.9b00576>). While the overall accuracy of the biocatalysis model is lower than that of the traditional one, the presented enzymatic models are sufficiently good for being used in daily research operations.. The larger data availability will increase the models' predictive power. In fact, the analysis of class and sub-class set sizes with accuracy (Figures S9 and S10) shows a growth in accuracy with a growth in data availability. Given the increasing use for enzymatic reactions in industry and the establishment of open data policies world-wide, we assume a steady growth of available data with important repercussion on the accuracy of the models in the near future.

Added / changed in conclusion: "The primary limitation of the data set, and thus of the model, is the scarcity of samples for some classes, which results in an imbalanced data set. However, as interest in biocatalysis grows and the research community embraces open data, we anticipate an increase in the quantity and quality of available training data, as well as experimental validation of proposed synthetic routes."

Page 18, lines 11-16

17. *There is a focus on biocatalysis in this work, when perhaps biosynthesis should be discussed given that the dataset comes from a number of databases detailing reactions in metabolism.*

Answer: Biosynthesis is out of the scope for both the publication and our expertise. However, we hope by making our method and data public, and by publishing in a journal targeting a general audience, that it will be used by researchers in adjacent fields.

18. *Is there scope to use the retrosynthesis tool for metabolic engineering?*

Answer: As we do not have the domain expertise to comment on this in our manuscript, we certainly see the potential given the nature of our training data.

19. *I imagine the contents of the dataset constructed in this work vs Chem. Sci., 2021, 12, 8648 is quite different, as this work takes reactions from BRENDA, Rhea, PathBank and MetaNetX, whilst the referenced work takes reactions from Reaxys. The reaxys dataset is likely more suitable for making suggestions for the use of enzymes for organic synthesis (biocatalysis), whilst the dataset constructed by the authors is likely more suited to biosynthesis or metabolic engineering? This difference in the datasets could be better explored in the manuscript?*

Answer: As the data from Reaxys is proprietary and not publicly available, it is not possible for us to make a comparison, especially when publishers (like Nature) call for open data policies to enforce reproducibility. This would also mean that it would not be possible for us to release all the data discussed in our manuscript and likely not being able to submit the work according FAIR standards.

20. *Following this point, how well can the model be expected to generalise to synthetic substrates if the dataset is limited to only the natural substrates of enzymes, even if/as this dataset grows.*

Answer: As we discuss in the section Attention Analysis and show in Figure 12, the model is capable to generalize using the attention mechanism. This is also shown by the fact that the model learns how to generate molecules mainly on the USPTO data set, which mainly contains synthetic substrates, while performing well, as shown in this study, on the substrates of enzymes.

21. *What is the advantage of the approach described by the authors over that adopted by RetroRules, which utilises a similar dataset. (<https://retrorules.org/>) (<https://doi.org/10.1093/nar/gky940>). Perhaps a comparison to RetroPath would be worth including?*

Answer: We would like to thank the reviewer for bringing RetroRules to our attention. We added RetroRules to our introduction as an example of rule-based approaches and cite it accordingly.

Changed to: "Currently, only rule-based methods for predicting biosynthesis pathways have been examined, such as the ATLAS of Biochemistry or RetroRules."

Page 3, lines 10-12

Reviewer 3

1. *However, EC numbers might not be available when considering reactions like those found in the USPTO data set, which method are proposing the authors in order to process such data set? The answer is not completely clear in the manuscript.*

Answer: Given that the models we developed are trained via multi-task transfer learning on a combination of enzymatic (w/ EC) and organic (w/o EC), they can naturally deal with missing EC information.

2. *The main purpose of this study seems to be a comparison with the RetroBioCat retrosynthesis tool as a benchmark, although it is not clear if the proposed tool outperforms RetroBioCat.*

Answer: The main purpose of this study is to create a data-driven alternative to the rule-based RetroBioCat. In addition, we show the first purely data-driven method for retrosynthetic pathway prediction. We have updated the abstract and introduction to better clarify this aspect of novelty. Assessing the performance relative to RetroBioCat in a quantitative way is outside of the scope of this work as data required to do so has not been made publicly available by RetroBioCat authors.

Update in the Abstract: "As of now, only rule-based systems support retrosynthetic planning using biocatalysis, while initial data-driven approaches are limited to forward predictions. Here, we extend the data-driven forward reaction as well as retrosynthetic pathway prediction models based on the Molecular Transformer architecture to biocatalysis."

Page 1, lines 7-10

Added to the introduction: "In addition, the lack of a corresponding backward model in this work does not allow for retrosynthetic planning."

Page 4, lines 16-17

3. *One issue is that even if accuracies seem to be higher than in the original proposed Molecular Transformer, they are still too low for cases that are well known to be hard to predict, such as for isomerases (EC 5). Therefore, there is no substantial progress from the new tool in that direction.*

Answer: We addressed the issues regarding discrepancies between enzyme classes by assessing them separately and show that these are caused by availability of data.

We show that our data-driven approach learns reaction rules that in previous works has do be defined manually (see Figures 4 and S17). This, as discussed in the introduction and conclusion, allows us to scale upon emergence of new data without additional work (manual curation of reaction rules).

We think that, in part, it is this data-driven approach that contributes to the fields of biocatalysis and chemistry more generally.

4. *The writing is at some points awkward, for instance trivial sentences like "enzyme-catalysed reactions are usually performed in water..." or "the reaction classes' statistics of Finnigan". Moreover, the Github link is not working.*

Answer: Due to a lack of specificity in this comment, we cannot conceive any required changes to the manuscript. The Github repository will be set to public upon publication, as stated in the manuscript.

REVIEWER COMMENTS

Reviewer #1 (Remarks to the Author):

I am happy with the corrections made and I am now able to recommend publication.

Reviewer #2 (Remarks to the Author):

I would like to see this work published, but I still have two big concerns which I feel the authors haven't addressed adequately.

1.

In my opinion the inclusion of analysis of data in the methods section is not acceptable. Large parts of the methods read as a second results and discussion section. I made this point previously (point 8, 9 and 15) but the authors have opted not to address it, claiming these results are a 'byproduct' of the main study.

In my opinion the best solution for this would be to include some of the key points at the beginning of the results chapter, to introduce the dataset and discuss its limitations. I note reviewer 1 makes a similar suggestion at point 6. However, if the rest of the results/discussion which is currently presented in the materials and methods is not interesting enough for the main paper, this should surely go in the supporting information?

I hope I am not being over-zealous to what a methods section should be. Perhaps the editor has an opinion on this?

2.

I raised a number of points (17-21 in the response) about the difference between datasets from biocatalysis and biosynthesis, which are not addressed in the paper. In particular, the dataset used by the authors is very much a biosynthesis dataset / a dataset containing mostly metabolic reactions, in stark contrast to the dataset from Reaxys used by the Raymond group earlier this year.

I appreciate that the code developed by the authors could be utilised with a completely different dataset, and they wish to present the code/approach as the focus of the study. However, in the study the results are completely dependent on the specific dataset used. The fact that the results of the retrosynthesis offer complementary suggestions to those made by RetroBioCat is driven in large part by the dataset chosen. For example if the authors were to use the Reaxys dataset, the retrosynthesis results would likely be quite different and possibly more similar to the suggestions made by RetroBioCat. (I don't think a quantitative comparison is necessary to comment on this, point 19). Furthermore, the dataset the authors have constructed is not likely to contain that many examples of enzymes working on synthetic substrates (ie biocatalysis). This is a separate issue to the problems with imbalanced data, and will limit suggestions for synthetic substrates (points 13, 14 and 20).

I find it surprising that the authors choose not to address or discuss this limitation, particularly as the paper is specifically framed around synthesis planning for biocatalysis.

Indeed, the lack of a good biocatalysis database is an issue for the whole field.

Other reviewers points.

Reviewer 3 – point 2.

The authors make a rebuttal that "Assessing the performance relative to RetroBioCat in a quantitative way is outside of the scope of this work as data required to do so has not been made publicly available by RetroBioCat authors". In the RetroBioCat paper a list of pathway rankings without the use of a database of literature reactions is presented. Since the RetroBioCat rules are manually created, there isn't any data which could be provided by the RetroBioCat authors which would be necessary to make a comparison with

those results?

However I think this is beside the point. The authors already show that their tool comes up with alternative suggestions which might be complementary to the suggestions from RetroBioCat. I think this is very much related to the point I make about the dataset above, which as I've said, should be better addressed.

Reviewer #3 (Remarks to the Author):

The authors have made some changes in the revision, but they seem to be just a rewriting of the same manuscript.

I do not understand why the authors have cut the comments from the reviewers and dismissed parts of them in the rebuttal letter. I do not think that this is the correct way of replying to the comments.

Moreover, I have not been able to find a version of the manuscript with highlighted changes.

Regarding the reply to the selected sections of my comments, they are not satisfactory at all. The authors seem to use as only argument that their approach is "data-driven", but they fail to answer my question about isomerases (comment #3) or, even worse, my suggestions involving sentences that might require revision (comment #4) are ignored while, surprisingly, the authors argue that my comment lacks specificity.

We would like to express our gratitude to the referees for carefully reading our manuscript and providing constructive feedback that significantly improved the paper's quality. We considered all the points raised. The referee's suggestions regarding the text and grammar have been incorporated into the revised version. In the following, we report a detailed response for each of the feedback: the referee's comments are repeated (*enumerated and written in italic*), followed by a detailed response by the authors prefixed with "Answer:" and the amended text in the article when that was needed.

Reviewer 1

1. *My comments are based not only on the machine learning merit of the study but also from a medicinal and organic chemistry vantage point. While the machine-learning model itself is not new (this is clearly stated by the authors)*

Answer: We would like to thank referee #1 for the positive feedback. The architecture of the model was initially published in 2019 by the IBM team and it has been used before for the prediction of traditional organic chemistry reactions (see the several group's publications) and, more recently, for forward enzymatic reaction prediction (Kreutter et al., 2021). However, the training of a model for backwards / retro predictions is a completely new use of this architecture in the context of enzymatic reaction. Moreover, unlike the work of Kreutter et al., 2021, we are encoding enzymatic reaction data using the EC number to make a better use of all those chemical reaction records that have little statistical relevance. We modified the abstract and the introduction to convey this message more clearly

2. *The work is very well written but, at times, a bit too overwhelming for a more general chemistry audience, in particular experimentalists.*

Answer: We tried to strike a balance between describing the work for a general chemistry experience and providing sufficient technical details to support reproducibility by other groups/researchers. Moreover, we made the entire work more easily consumable by giving access to the deployed model on a graphical user interface to lower the adoption barrier for non-ML experts in their daily research work. Some of the next points have been carefully considered to make the work less overwhelming and more easily readable by a larger audience.

3. *Why did the authors choose a model trained on the USPTO dataset as baseline? How is it meaningful for the research question in this study?*

Answer: The volume of enzymatic reaction data alone is insufficient (70K entries) to ensure proper SMILES grammar learning. The larger USPTO data set is required in addition to the enzymatic reaction data for the model to learn the SMILES grammar using multi-task transfer learning. This approach has been already validated by Pesciullesi et al. (<https://www.nature.com/articles/s41467-020-18671-7>), who demonstrated the use of transfer learning when training predictive models with small data sets.

Added: "The USPTO data is used to learn general chemistry knowledge and the SMILES grammar, as the comparatively small enzymatic reaction data set does not provide sufficient information for these tasks."

Page 20, lines 5-7

4. *"The forward prediction model achieves an accuracy of 49.6% and 62.7%, top-1 and top-5 respectively, while the single-step retrosynthetic model shows a round-trip accuracy of 39.6% and 42.6%, top 1 and top-10, respectively" For consistency, I would suggest presenting the top-1/5 or top1/10 in both cases.*

Changed to: "The forward prediction model achieves an accuracy of 49.6%, 63.5%, and 68.8% top-1 and top-5, and top-10 respectively, while the single-step retrosynthetic model shows a round-trip accuracy of 39.6%, 42.3%, and 42.6%, top-1, top-5, and top-10, respectively."

Page 5, lines 1-3

5. *"An extensive analysis of the data set pinpoints the performance to the enzyme class distribution of the training data, with the forward prediction model ranging from an accuracy of 18.6% and round-trip accuracy of 1.7% for isomerases, the most scarce class, to a forward prediction accuracy of 64.4% and a round-trip prediction accuracy of 60.5% for transferases, the most abundant class." The sentence is a bit convoluted. I would suggest rephrasing or dividing it in two.*

Answer: We have removed this sentence from the introduction based on comments by reviewer 2. Their reasoning, which we agree with, is that it belongs in the results section.

6. *The manuscript would benefit from Figure 1 demonstrating the data distribution across different classes. I think Figure 10 is overwhelming. Maybe the middle pie chart is enough for the main manuscript. The caption would then link to the full Figure 10 (moved to SI) and Tables S3-4 for details.*

Answer: We believe that the reference to Figure 10 was referring to Figure 2. We transferred the entire figure to the SI (now Figure S3) while keeping only the middle section in the main manuscript as Figure 2.

7. *I think that top-k accuracy is not the best way of assessing models, but in all honesty I am not able to suggest a viable and validated alternative that is simultaneously convincing for a cheminformatics and medicinal chemist/chemical biologist audience. There are however caveats to top-k metrics and maybe the authors could discuss them a bit in the manuscript (a few sentences would be enough) for better context and targeted at a more general audience.*

Answer: We don't fully understand the general comment about the top-k metric's caveats. We can confirm that, while not perfect, top-k accuracy emerged as the most commonly used metric to evaluate comparable tasks (e.g.

<https://pubs.rsc.org/en/content/articlelanding/2021/sc/d1sc02362d>,

<https://pubs.rsc.org/en/content/articlehtml/2020/sc/c9sc03666k>).

8. *The authors point towards a limitation on the study. The database does not seem to be homogeneous in several aspects that are key for a correct comparison between predictions and ground truth. Although this does not invalidate the conclusions, I would suggest introducing a word of caution or (ideally) a section regarding limitations in the dataset. This would probably mean de-emphasizing the top-k values discussion in the manuscript to a certain extent, since those limitations are likely skewing the performance evaluation. If they are not skewing, which data curation measures were taken? How many database entries have issues in the stereochemistry (unassigned, non-conclusive, wrong) protonation states and/or others (either on substrate or product sides)? I reckon this is an important point so that each reader can make an informed and independent assessment on the value of the database.*

Answer: The limitations of the data set do indeed skew the performance evaluation. We identified the uneven distribution of training samples across classes and sub-classes as the primary issue with the data set and addressed it through a split analysis (see Figures 5 and 8, as well as Figures

S9 and S10). The questions about stereochemical issues, protonation states, and so on can only be addressed by manual curation strategies that are human intensive and cannot be dealt with by any existing automatism. We used the available dataset as ground truth because data curation is beyond the scope of this work. In addition, we made the entire dataset available to others so that they could use and analyse it more thoroughly and perhaps even improving its quality in the future.

Having said that, it is worth noting that we were able to demonstrate that both the forward and backwards models are resilient to errors in the ground truth (Figures 7 and 11, respectively). We added the following text to the conclusion to highlight the data set's limitations.

Added: "The primary limitation of the data set, and thus of the model, is the scarcity of samples for some classes, which results in an imbalanced data set. However, as interest in biocatalysis grows and the scientific community embraces open data rules, we predict an increase in the quantity and quality of available training data, as well as experimental validation of proposed synthetic routes. "

Page 18, lines 11-16

9. *Figure 3: rather than incorrect predictions, the figure should be captioned with something else because most "incorrect" predictions are indeed deficiencies in the source data, e.g. the ground truth for reaction 3 (Figure 3) has two errors.*

Answer: This is indeed true, and we changed the captions in Figures 7 and 11 accordingly. We chose a more neutral caption as shown below.

Changed to: "Inspection of forward predictions labelled as incorrect." and "Inspection of backward predictions labelled as incorrect."

Page 8, line 1 and page 13, line 1

10. *I do not quite follow the justification for reaction 5. Maybe depict the reaction the authors allude to.*

Answer: We hope that the updates shown below make the issue clear. The reaction is, in fact, depicted as the ground truth, and the correct reaction / product (according to the test set) in black (substrate + enzyme) and red (product).

X | 1.2.3.4 >> Y
X | 1.2.3.6 >> Z

Discarding EC-level 4 of the two reactions shown above leads to:

X | 1.2.3 >> Y (this reaction was part of the training set)
X | 1.2.3 >> Z (this reaction was part of the test set)

Changed to: "Concerning the diester hydrolase-catalyzed reaction (4) and the intramolecular lyase reaction (5), it is worth noting that the data set contains few chemical reaction records with identical substrates and EC numbers that result in different products. With sufficient data, the model would be able to recommend, given an EC-number, the various transformations of a single substrate into different products with corresponding confidence levels. However, due to the limited data volume and the random nature of the split into training, validation, and test sets, it is unlikely that all possible outcomes of a single substrate and EC number will be used for training. As a result, the two reactions are predicted incorrectly."

Page 6, lines 18-22 and page 7, lines 1-4

11. *Typo in page 11, line 14: "int".*

Changed to: "low prediction accuracy in the class of oxidoreductases"

Page 10, line 13

12. *With the exception of route (5), which is highly substrate-specific, one can appreciate the dissimilarity of the synthetic pathways suggested by our model when compared with RetroBioCat, which opens the way to synthetically useful compounds from a variety of different inexpensive substrates." This reads a bit too assertive without experimental evaluation for proof-of-concept. I would suggest either rephrasing or adding an additional sentence saying these are still predictions and would warrant experimental work to more accurately assess the value of the recommended pathways.*

This is true. We changed the sentence as shown below to make it less assertive.

Changed to: "which opens a way to synthetically useful compounds from a variety of different inexpensive substrates once experimentally validated."

Page 16, line 11 and Page 17, lines 1-2

Reviewer 2

1. *The main difference being the use of EC number rather than enzyme name as a reaction token, and the construction of an alternative enzyme dataset.*

Answer: We would like to thank reviewer 2 for the constructive feedback. We would like to stress that the construction of an alternative enzyme dataset is a by-product and not the main scope of this work. Beyond using EC numbers as reaction token to take advantage of those reaction records with little statistical significance, the main novelty of the current work is the use of the molecular transformer for backward prediction to construct retrosynthetic pathways. We adapted the abstract to reflect this fact better.

2. *However, the work presented does not seem a particularly big step beyond previously published work (Chem. Sci., 2021, 12, 8648), and I wonder whether the accuracies presented mean this is unlikely to be taken up by practitioners of biocatalysis or metabolic engineering?*

Answer: We stress again that prior art using Molecular Transformer was only capable of predicting forward reaction outcomes and could not be used to construct retrosynthetic pathways. We adapted the abstract and introduction to make the contribution to the field of biocatalysis clearer.

3. *I would expect a few more citations in the opening paragraph of the introduction to support some of the statements made there.*

Answer: We added the following additional citations to support the statements in the opening paragraph:

- Trewavas, A. Malthus Foiled Again and Again. *Nature* **2002**, *418* (6898), 668–670. <https://doi.org/10.1038/nature01013>.
- Matlin, S. A.; Abegaz, B. M. Chemistry for Development. In *The Chemical Element: Chemistry's Contribution to Our Global Future*; García-Martínez, J., Serrano-Torregrosa, E., Eds.; Wiley-VCH, **2011**; pp 1–70.
- Duigou, T.; du Lac, M.; Carbonell, P.; Faulon, J.-L. RetroRules: A Database of Reaction Rules for Engineering Biology. *Nucleic Acids Research* **2018**, *47* (D1), D1229–D1235. <https://doi.org/10.1093/nar/gky940>.
- Mazurenko, S.; Prokop, Z.; Damborsky, J. Machine Learning in Enzyme Engineering. *ACS Catalysis* **2020**, *10* (2), 1210–1223. <https://doi.org/10.1021/acscatal.9b04321>.

Page 2, lines 2-9

4. *Page 2 line 19 – “enzymatic reactions to catalyse organic synthesis”. Small thing, but it’s the enzyme (not the reaction) that is the catalyst. Just needs re-wording.*

Changed to: "Although the ability to use enzymes as catalysts in the organic synthesis of chemical compounds gained widespread attention for large scale production"

Page 2, lines 19-21

5. *Page 3 line 10 – “One of the first work” – the sentence reads strangely and might want to be re-written.*

Changed to: "Currently, only rule-based methods for predicting biosynthesis pathways have been examined, such as the ATLAS of Biochemistry or RetroRules."

Page 3, lines 10-12

6. *Page 4 line 19-20 – “The use of a backward model allows to predict substrates and catalysing enzyme classes given a target product.” - This also reads badly*

Answer: We changed the text, and based on your previous comments, also added emphasis on the differentiation to previous work to clarify the novelty of our data-driven backward model.

Changed to: "Compared to the previous work by Kreutter et al., a backward model allows to predict substrates and enzyme classes given a target product, enabling retrosynthetic pathway prediction. In addition, we incorporate the EC (enzyme commission) number into the reaction SMILES, rather than encoding enzymes with their natural language name."

Page 4, lines 20-24

7. *Page 4 line 17 onwards – Large parts of this feel like it should be in the results. This paragraph seems to go beyond introducing the aims and objectives of the work and into describing specifically what was done. I would have thought information about constructing the ECREACT dataset should be in the results rather than the introduction?*

Answer: As this is already being discussed extensively in the Results and Discussion and Methods sections (see further answers), we have shortened the discussion on the data set in the introduction.

Removed: “The resulting data set contains more than 62,000 unique enzymatic reactions.”

Removed: “The USPTO data set contains 1 million reactions without enzymatic information. These reactions acted as a training set for learning the general knowledge of chemical reactions and the SMILES grammar.”

Removed: “An extensive analysis of the data set establishes a connection between performance and the enzyme class distribution of the training data. For the most scarce class, the isomerases, the forward prediction accuracy is 18.6% and the round-trip accuracy 1.7%, while for the most abundant class, the transferases, the forward prediction accuracy is 64.4% and the round-trip accuracy 60.5%.”

Removed: “The forward and backward (substrate + EC -> product and product -> substrate + EC, respectively) models were then trained using multitask transfer learning on the ECREACT and the USPTO data sets (see Data Sets and Model Training).”

8. *Page 5 line 15 – The EC3 data is presented without any information about what this is or where it's come from. In the materials and methods this is presented. Maybe it would be useful to start the results section with a brief description of the construction of this dataset (rather than talking about this in the introduction), and specifically what the EC3 data is. This seems an important part of the narrative before we get into the forward or reverse prediction?*

Answer: While we think the data set will provide useful to the field of machine learning for biocatalysis, it is not the main result of the study. To improve readability and wider audience interest, we prefer to refer the reader to the Methods section. We changed the text with a reference to the appropriate section for the reader to learn more about the data set.

Changed to: "The Dataset was constructed following the details reported in the Methods, Data Sets and Model Training. We split the EC3 data set (n=56579) into a test and a training set, enforcing a zero overlap between the product distributions of the two ensembles, i.e. no product molecule present in the test set appears in the training set."

Page 5, lines 6-9

9. *It does seem that a number of results and discussion are currently presented in the materials and methods section, which doesn't seem correct. I would have thought the construction and analysis of the ECRReact dataset, and the analysis around the different EC levels belongs in the results and discussion section. Furthermore, figures 10 and 11 are clearly an analysis of the dataset rather than a method.*

Answer: We do not consider the dataset the main outcome of this work, but rather an important byproduct that will help stimulate the use of machine learning in biocatalysis.

10. *Page 11 line 5 – Figure 12 Shouldn't this figure simply be the next numbered figure in the results section? (so figure 12 should actually be figure 5?)*

Answer: Indeed. As the sentence becomes superfluous with the changes, we removed it entirely.

Removed: Figure 12 shows the backward model performance (round-trip and top-k accuracy) for the EC-level 3.

11. *Figure 8 – are the retrobiocat reactions similar to each other simply because they are in cascades, so the reaction participants are similar? Taking metabolic pathways from BRENDA or Rhea, would a similar result be achieved compared the entire ECRReact dataset?*

Answer: The reactions from RetroBioCat are clustered together because they have similar reaction mechanisms and do not show cascades of metabolic pathways.

12. *I think the point raised about the differences between the routes suggested by RetroBioCat and the transformer presented here is an interesting one. Is there a tendency of the model presented to suggest routes which might be more closely aligned with metabolism, given that all the training data is for reactions from metabolism?*

Answer: Naturally, a data-driven model inherits biases present in the training data set. ECREACT's sources are skewed towards metabolic reactions and this is reflected in the suggestions of our trained models.

13. *How similar are the reactions suggested by the retrosynthesis tool to reactions in the dataset? Are these reactions in the dataset?*

Answer: They are similar to ones in the training set (ML) as shown in Figure 8.

14. *Do the reactions suggested by the tool have an advantage over the retrobiocat suggestions in that they are likely to be similar to natural substrates, with a reasonable likelihood of being accepted (as opposed to the more general rules used by retrobiocat)? However possibly the scope for what will be suggested is smaller.*

Answer: When comparing to RetroBioCat, we encountered the challenge of not having access to all of the data used to build RetroBioCat. RetroBioCat, to our surprise, only shares a small set of data, not the entire set used to construct the chemical reaction rules. This precludes more thorough analysis and comparison. Our model's recommendations reflect the training data, which was largely derived from reactions containing natural substrates. However, the Molecular Transformer can learn the chemical transformation rules (for a proof, see <https://www.science.org/doi/10.1126/sciadv.abe4166>), and we show in this paper (see Figure 12 and section "Attention Analysis") that the model can learn the importance of the EC token in relation to the reaction centres. As a result, the trained model can successfully extend the learned biochemical transformation rules to chemical structures other than natural products.

15. *As mentioned, there are results and discussion presented here which should be moved to provide a better stand-alone narrative in the actual results and discussion section*

Answer: See answer to comment 14.

16. *Are the accuracies presented too low to be of much practical use? Perhaps this could change with a larger dataset? How much data do the authors envisage needing to achieve reliable predictions, and is this feasible in the medium-near future? This seems a major part of the conclusions presented by the authors. How rapidly is the available data increasing year on year?*

Answer: Our previous publication on (traditional organic chemistry) backward / retro reaction prediction has gained a lot of attention by the entire chemistry community (<https://pubs.acs.org/doi/full/10.1021/acscentsci.9b00576>). While the overall accuracy of the biocatalysis model is lower than that of the traditional one, the presented enzymatic models are sufficiently good for being used in daily research operations.. The larger data availability will increase the models' predictive power. In fact, the analysis of class and sub-class set sizes with accuracy (Figures S9 and S10) shows a growth in accuracy with a growth in data availability. Given the increasing use for enzymatic reactions in industry and the establishment of open data policies world-wide, we assume a steady growth of available data with important repercussion on the accuracy of the models in the near future.

Added / changed in conclusion: "The primary limitation of the data set, and thus of the model, is the scarcity of samples for some classes, which results in an imbalanced data set. However, as interest in biocatalysis grows and the research community embraces open data, we anticipate an increase in the quantity and quality of available training data, as well as experimental validation of proposed synthetic routes."

Page 18, lines 11-16

17. *There is a focus on biocatalysis in this work, when perhaps biosynthesis should be discussed given that the dataset comes from a number of databases detailing reactions in metabolism.*

Answer: Biosynthesis is out of the scope for both the publication and our expertise. However, we hope by making our method and data public, and by publishing in a journal targeting a general audience, that it will be used by researchers in adjacent fields.

18. *Is there scope to use the retrosynthesis tool for metabolic engineering?*

Answer: As we do not have the domain expertise to comment on this in our manuscript, we certainly see the potential given the nature of our training data.

19. *I imagine the contents of the dataset constructed in this work vs Chem. Sci., 2021, 12, 8648 is quite different, as this work takes reactions from BRENDA, Rhea, PathBank and MetaNetX, whilst the referenced work takes reactions from Reaxys. The reaxys dataset is likely more suitable for making suggestions for the use of enzymes for organic synthesis (biocatalysis), whilst the dataset constructed by the authors is likely more suited to biosynthesis or metabolic engineering? This difference in the datasets could be better explored in the manuscript?*

Answer: As the data from Reaxys is proprietary and not publicly available, it is not possible for us to make a comparison, especially when publishers (like Nature) call for open data policies to enforce reproducibility. This would also mean that it would not be possible for us to release all the data discussed in our manuscript and likely not being able to submit the work according FAIR standards.

20. *Following this point, how well can the model be expected to generalise to synthetic substrates if the dataset is limited to only the natural substrates of enzymes, even if/as this dataset grows.*

Answer: As we discuss in the section Attention Analysis and show in Figure 12, the model is capable to generalize using the attention mechanism. This is also shown by the fact that the model learns how to generate molecules mainly on the USPTO data set, which mainly contains synthetic substrates, while performing well, as shown in this study, on the substrates of enzymes.

21. *What is the advantage of the approach described by the authors over that adopted by RetroRules, which utilises a similar dataset. (<https://retrorules.org/>) (<https://doi.org/10.1093/nar/gky940>). Perhaps a comparison to RetroPath would be worth including?*

Answer: We would like to thank the reviewer for bringing RetroRules to our attention. We added RetroRules to our introduction as an example of rule-based approaches and cite it accordingly.

Changed to: "Currently, only rule-based methods for predicting biosynthesis pathways have been examined, such as the ATLAS of Biochemistry or RetroRules."

Page 3, lines 10-12

Reviewer 3

1. *However, EC numbers might not be available when considering reactions like those found in the USPTO data set, which method are proposing the authors in order to process such data set? The answer is not completely clear in the manuscript.*

Answer: Given that the models we developed are trained via multi-task transfer learning on a combination of enzymatic (w/ EC) and organic (w/o EC), they can naturally deal with missing EC information.

2. *The main purpose of this study seems to be a comparison with the RetroBioCat retrosynthesis tool as a benchmark, although it is not clear if the proposed tool outperforms RetroBioCat.*

Answer: The main purpose of this study is to create a data-driven alternative to the rule-based RetroBioCat. In addition, we show the first purely data-driven method for retrosynthetic pathway prediction. We have updated the abstract and introduction to better clarify this aspect of novelty. Assessing the performance relative to RetroBioCat in a quantitative way is outside of the scope of this work as data required to do so has not been made publicly available by RetroBioCat authors.

Update in the Abstract: "As of now, only rule-based systems support retrosynthetic planning using biocatalysis, while initial data-driven approaches are limited to forward predictions. Here, we extend the data-driven forward reaction as well as retrosynthetic pathway prediction models based on the Molecular Transformer architecture to biocatalysis."

Page 1, lines 7-10

Added to the introduction: "In addition, the lack of a corresponding backward model in this work does not allow for retrosynthetic planning."

Page 4, lines 16-17

3. *One issue is that even if accuracies seem to be higher than in the original proposed Molecular Transformer, they are still too low for cases that are well known to be hard to predict, such as for isomerases (EC 5). Therefore, there is no substantial progress from the new tool in that direction.*

Answer: We addressed the issues regarding discrepancies between enzyme classes by assessing them separately and show that these are caused by availability of data.

We show that our data-driven approach learns reaction rules that in previous works has do be defined manually (see Figures 4 and S17). This, as discussed in the introduction and conclusion, allows us to scale upon emergence of new data without additional work (manual curation of reaction rules).

We think that, in part, it is this data-driven approach that contributes to the fields of biocatalysis and chemistry more generally.

4. *The writing is at some points awkward, for instance trivial sentences like "enzyme-catalysed reactions are usually performed in water..." or "the reaction classes' statistics of Finnigan". Moreover, the Github link is not working.*

Answer: Due to a lack of specificity in this comment, we cannot conceive any required changes to the manuscript. The Github repository will be set to public upon publication, as stated in the manuscript.

REVIEWERS' COMMENTS

Reviewer #2 (Remarks to the Author):

The authors have answered all my comments, and I can now recommend publication provided reviewer 3 is happy with the response to their point 2.

Point 1.

The inclusion of a section detailing the construction of the dataset at the start of the results-discussion is a welcome addition. In my opinion this makes the narrative of the result-discussion section much easier to follow. The methods section seems much better.

Point 2.

The new paragraphs are an excellent addition to the paper.

Reviewer 3 – point 2.

What do the authors mean that the data to produce the expertly encoded reactions rules is not available? There is no dataset used in the construction of manually curated rules, just expert knowledge and the literature in general. The authors have access to this information, and to the rules which were manually curated if they required them (https://figshare.com/articles/software/RetroBioCat_database_files/12696482).

At the end of the day, if the authors wanted to make a quantitative comparison of RetroBioCat vs their transformer approach trained on the ECREACT data, this is easily achievable. Simply, can the transformer suggest the cascades listed in the RetroBioCat paper, and if so, how highly does it rank them in comparison to RetroBioCat. Actually, no data other than the RetroBioCat paper is required for this. Furthermore, it would also be possible to run the RetroBioCat code in the mode which ignores literature precedent, if the authors needed to re-run these experiments. Whilst the RetroBioCat authors haven't made their complete dataset for literature precedent reactions public, this is not necessary for a comparison.

I am therefore unconvinced by the authors argument that a direct comparison is impossible.

However, as I mentioned in my last comment, in my opinion a direct comparison is not necessary. The authors have done a good job at highlighting that their tool comes up with alternative suggestions, likely due to the dataset used. Furthermore their tool can be trained on new datasets, which RetroBioCat can not. I do not find the lack of a direct comparison a problem (despite my disagreement about how possible it is to do this).

Reviewer #3 (Remarks to the Author):

In this revised version of the manuscript, the authors have addressed my comments in a more systematic and consistent way, compared with the previous version. Such effort from the authors has allowed me to assess more objectively their work.

I understand the issue that the RetroBioCat and Reaxys data are not publicly available and, thus, the associated limitations for the authors in terms of validation of the proposed method.

Overall, the work brings commensurable news insights into biocatalysed synthesis planning and contributes reasonably one step further to the state of the art in the field.

Therefore, I recommend publication of this work.

Answers to Reviewers

Reviewer #2 (Remarks to the Author)

The authors have answered all my comments, and I can now recommend publication provided reviewer 3 is happy with the response to their point 2.

Point 1.

The inclusion of a section detailing the construction of the dataset at the start of the results-discussion is a welcome addition. In my opinion this makes the narrative of the result-discussion section much easier to follow. The methods section seems much better.

Point 2.

The new paragraphs are an excellent addition to the paper.

Answer (Point 1 and 2): We would like to thank the referee for the comments and suggestions that improved the quality of the paper and increased its accessibility and value to the readership.

Reviewer 3 – point 2.

What do the authors mean that the data to produce the expertly encoded reactions rules is not available? There is no dataset used in the construction of manually curated rules, just expert knowledge and the literature in general. The authors have access to this information, and to the rules which were manually curated if they required them (https://figshare.com/articles/software/RetroBioCat_database_files/12696482).

At the end of the day, if the authors wanted to make a quantitative comparison of RetroBioCat vs their transformer approach trained on the ECREACT data, this is easily achievable. Simply, can the transformer suggest the cascades listed in the RetroBioCat paper, and if so, how highly does it rank them in comparison to RetroBioCat. Actually, no data other than the RetroBioCat paper is required for this. Furthermore, it would also be possible to run the RetroBioCat code in the mode which ignores literature precedent, if the authors needed to re-run these experiments. Whilst the RetroBioCat authors haven't made their complete dataset for literature precedent reactions public, this is not necessary for a comparison.

I am therefore unconvinced by the authors argument that a direct comparison is impossible.

However, as I mentioned in my last comment, in my opinion a direct comparison is not necessary. The authors have done a good job at highlighting that their tool comes up with alternative suggestions, likely due to the dataset used. Furthermore their tool can be trained on new datasets, which RetroBioCat can not. I do not find the lack of a direct comparison a problem (despite my disagreement about how possible it is to do this).

Answer: We would like to thank the referee for their detailed response. As the referee highlights, the paper already provides some comparisons by highlighting alternative suggested routes. Based on the referee feedback, we agree that additional comparisons are not necessary.

In conclusion, we believe it is important to clarify our statements in the last reply about RetroBioCat data availability: Although reaction rules are based on “*expert knowledge and the literature in general*”, the literature reaction data (used in RetroBioCat) is essential for retraining our models and run a fair comparison between the two methods. While the derived rules are available for download, this data is not usable in ML training tasks. Instead, it is essential to have a corresponding set of literature reaction data to retrain our models. Unless this data is available (which is, to the best of our knowledge, not the case) we have no way of guaranteeing a fair comparison between the two approaches.

Reviewer #3 (Remarks to the Author)

In this revised version of the manuscript, the authors have addressed my comments in a more systematic and consistent way, compared with the previous version. Such effort from the authors has allowed me to assess more objectively their work.

I understand the issue that the RetroBioCat and Reaxys data are not publicly available and, thus, the associated limitations for the authors in terms of validation of the proposed method.

Overall, the work brings commensurable news insights into biocatalysed synthesis planning and contributes reasonably one step further to the state of the art in the field.

Therefore, I recommend publication of this work.

Answer: We would like to again thank the Reviewer for their work and help to substantially increase the quality and accessibility of the manuscript.